# Direct observation of the conformational states of PIEZO1

Eric M. Mulhall[1], Anant Gharpure[1], Rachel M. Lee[2], Adrienne E. Dubin[1], Jesse S. Aaron[2], Kara L. Marshall[1,5], Kathryn R. Spencer[3], Michael A. Reiche[2], Scott C. Henderson[4], Teng-Leong Chew[2] & Ardem Patapoutian[1✉]

PIEZOs are mechanosensitive ion channels that convert force into chemoelectric signals[1,2] and have essential roles in diverse physiological settings[3]. In vitro studies have proposed that PIEZO channels transduce mechanical force through the deformation of extensive blades of transmembrane domains emanating from a central ion-conducting pore[4-8]. However, little is known about how these channels interact with their native environment and which molecular movements underlie activation. Here we directly observe the conformational dynamics of the blades of individual PIEZO1 molecules in a cell using nanoscopic fluorescence imaging. Compared with previous structural models of PIEZO1, we show that the blades are significantly expanded at rest by the bending stress exerted by the plasma membrane. The degree of expansion varies dramatically along the length of the blade, where decreased binding strength between subdomains can explain increased flexibility of the distal blade. Using chemical and mechanical modulators of PIEZO1, we show that blade expansion and channel activation are correlated. Our findings begin to uncover how PIEZO1 is activated in a native environment. More generally, as we reliably detect conformational shifts of single nanometres from populations of channels, we expect that this approach will serve as a framework for the structural analysis of membrane proteins through nanoscopic imaging.

The capacity to sense and transduce mechanical information from the environment is critical to a wide variety of physiological processes across all domains of life[9]. Mechanotransduction channels harness mechanical work to directly open an ion-conducting pore in response to perturbations in the cell membrane, initiating cellular signalling[10]. PIEZOs are a family of mechanotransduction channels found across Eukarya[1,2] and mediate a vast array of physiological processes in mammals, including touch sensation[11], blood pressure control[12], vascular development[13,14], mechanical itch[15] and erythrocyte hydration state[16].

PIEZOs are large, homotrimeric membrane proteins structurally arranged as a triskelion[4,5,7] (Fig. 1a). Each protomer contacts at the central pore and cap domains near the C terminus and projects a blade of 36 transmembrane domains that extends both outward and upward towards the N terminus. Although only partial cryo-electron microscopy (cryo-EM) structures of PIEZO1 lacking approximately one third of the distal blades are available, the structural homologue PIEZO2 has been solved with the full complement of transmembrane domains[17]. In structural models and predictions, the blades form a bowl shape approximately 24 nm in diameter and 9 nm in depth, with a total projected area of approximately 450 nm[2]. The blades directly connect to the pore via an intracellular beam, suggesting that they are both levers that directly gate the channel and the primary sensors of mechanical force[6].

When reconstituted into artificial lipid bilayers, the non-planar shape of PIEZO1 is sufficiently rigid to bend the membrane around it, forming a dome[8,18]. However, this dome is also intrinsically deformable, a property probably controlled by the flexibility of the blades. Observations of the membrane dome in lipid vesicles and accompanying mathematical models indicate that in a planar lipid bilayer with no lateral tension or compressive forces, PIEZO1 should flatten relative to the detergent-solubilized state[4,8,18,19]. Such deformation is driven by the energetic cost to bend the membrane, in which the curved PIEZO protein and the planar lipid bilayer are in a state of mechanical equilibrium. Externally applied forces appear to further deform the PIEZO dome, flattening when tapped with an atomic force microscope in a supported planar lipid bilayer[8], and in a molecular dynamics simulation of bilayer expansion through lateral tension[20]. A partial cryo-EM structure of PIEZO1 solved outside-out in a 10-nm lipid vesicle also shows blade and beam deformation under very high bending forces[21]. Together, these data support the model that the blades are sufficiently flexible to bend upon membrane deformation and probably transduce force to gate the pore. However, the complexity and heterogeneity of cell membranes have required that these experiments be performed in highly purified systems, and methods for applying force to the channel lack direct physiological relevance. The extensive averaging required to assemble cryo-EM structural models also fails in many cases to resolve the potential breadth of conformational states, especially from relatively flexible protein domains.

[1]Howard Hughes Medical Institute, Department of Neuroscience, Dorris Neuroscience Center, Scripps Research, La Jolla, CA, USA. [2]Advanced Imaging Center, Howard Hughes Medical Institute Janelia Research Campus, Ashburn, VA, USA. [3]Department of Neuroscience, Dorris Neuroscience Center, Scripps Research, La Jolla, CA, USA. [4]Department of Molecular Medicine, Scripps Research, La Jolla, CA, USA. [5]Present address: Department of Neuroscience, Baylor College of Medicine, Houston, TX, USA. ✉e-mail: ardem@scripps.edu

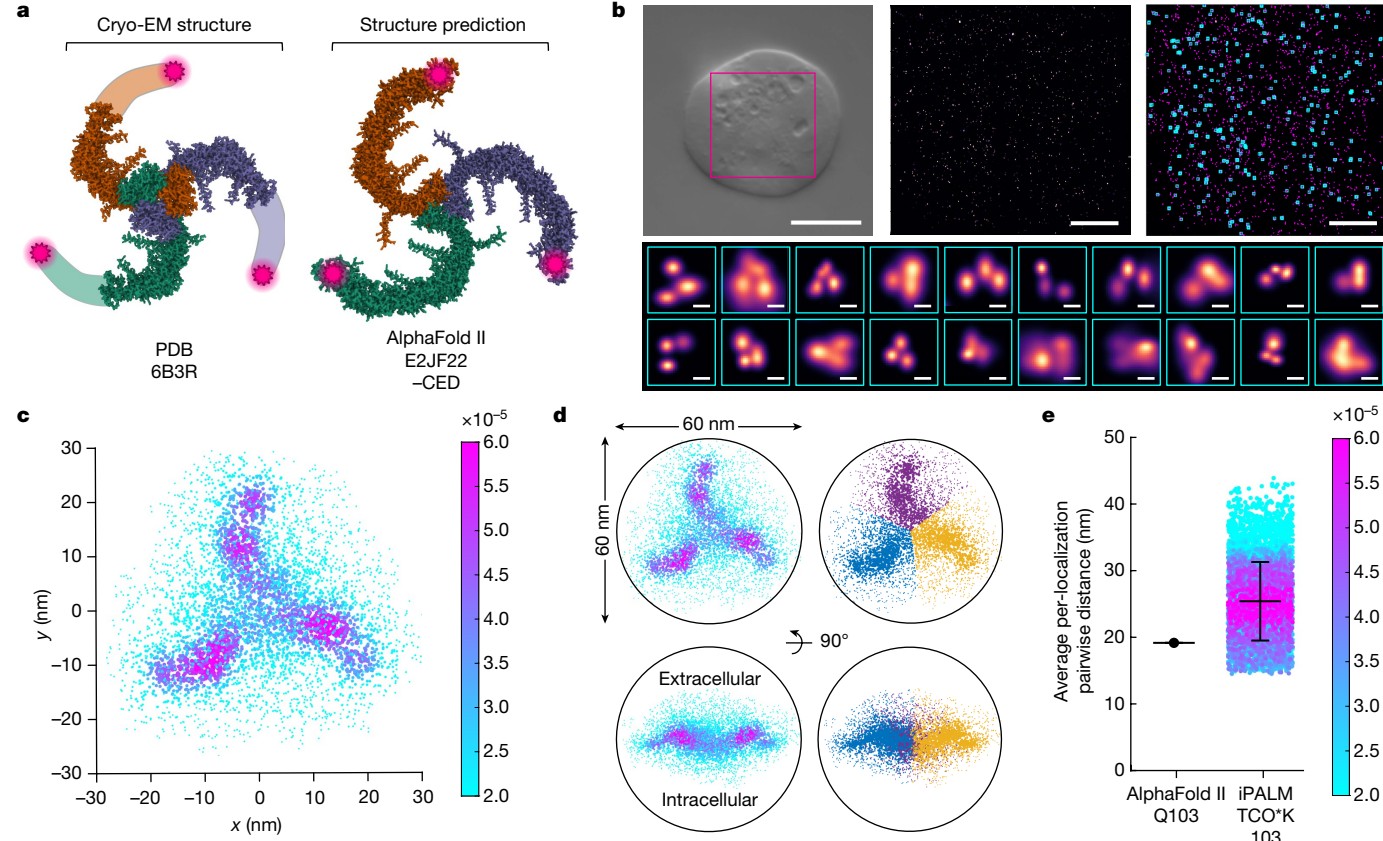

**Fig. 1 | Single-molecule imaging and super-particle fusion of labelled PIEZO1. a**, Structural models of PIEZO1, viewed extracellularly. The most complete cryo-EM structure of PIEZO1 with the missing distal approximately one third of the blade highlighted (left) and the AlphaFold II structure prediction of PIEZO1 (right) are shown. The C-terminal extracellular domain (CED) is removed from the AlphaFold II model due to poor prediction. TCO*K tags at position 103 are shown as magenta stars. Each protomer is separately coloured. **b**, Segmentation of candidate PIEZO1 particles from iPALM localizations. A representative ×100 differential interference contrast image (from n = 5 cells) of a HEK293 cell expressing TCO*K 103 PIEZO1 labelled with tetrazine–Alexa Fluor 647 (AF647) (top left; scale bar, 10 μm). A 3 nm per pixel rendering of plasma membrane localizations from the magenta inset on the top left (scale bar, 3 μm) (top middle). Binary AF647 localizations (magenta points) with candidate PIEZO1 molecules meeting nearest neighbour requirements highlighted with cyan boxes (scale bar, 3 μm) (top right). Representative 3 nm per pixel renderings of candidate triple-labelled PIEZO1 molecules that meet minimum interlocalization separation requirements are also shown (scale bars, 30 nm) (bottom). **c**, Fused super-particle of identified PIEZO1 trimeric localizations with threefold symmetry promotion, viewed top down (n = 5 imaged cells, n = 726 molecules and n = 8,500 localizations). Localizations were visualized with size and colour proportional to local density (see Methods). **d**, Super-particle thresholded for a local density of more than 0.5 × 10⁻⁵, the minimum that encompassed localizations within a 60-nm sphere (left). Localizations associated with each blade isolated with k-means clustering with k = 3 clusters (right). **e**, Scatter plot of average per-localization interblade distances between each localization within a blade cluster from the super-particle in part **c** and all localizations in a neighbouring blade cluster, coloured by local density. At position 103, the most probable interblade distance is 25.4 ± 5.9 nm (mean ± s.d.), compared with 19.2 nm calculated from the AlphaFold II model.

In this study, we overcome these previous limitations by using nanoscopic fluorescence imaging to directly observe the conformational states of the blades of single PIEZO1 channels in a cell membrane. Two super-resolution nanoscopic imaging approaches combined with novel particle identification and segmentation algorithms allowed us to isolate and measure single PIEZO1 molecules at nanometre resolution. Using this approach, we applied stimuli to cells and examined how blade expansion correlates with channel activation and inhibition. Through these experiments, we also begin to uncover the basic mechanism of channel modulation by the small-molecule activator Yoda1 and the inhibitor GsMTx-4. Together, our results show how the cellular environment can shape the structure of PIEZO1 and how blade expansion underlies channel activation.

## Single-molecule imaging of PIEZO1

Protein tags and affinity probes introduce error in super-resolution fluorescence microscopy as the fluorophore (or fluorophores) and tagged location are physically offset[22]. To minimize this type of spatial error, we labelled each subunit of PIEZO1 with a single extracellular fluorophore using genetic code expansion and click chemistry. An orthogonal aminoacyl–tRNA synthetase and tRNA pair recognizing the amber codon UAG[23] was used to incorporate a lysine conjugated to *trans*-cyclooctene (TCO*K) at amino acid position 103 in mouse PIEZO1 heterologously expressed in HEK293 cells. Although the last approximately one third of the PIEZO1 blade has not been resolved by cryo-EM[4,5,7], the structure of this region has been predicted by AlphaFold II[24] and homology modelling using the PIEZO2 structure[17] (Fig. 1a and Extended Data Fig. 1a). Amino acid 103 resides in the most distal extracellular loop of PIEZO1 relative to the pore, and these positions are separated by 19.2 nm in the tertiary structure (Extended Data Fig. 1b). Live cells expressing TCO*K 103 PIEZO1 were labelled with the complementary click substrate tetrazine conjugated to Alexa Fluor 647 (Extended Data Fig. 2a,b). This loop was confirmed to be extracellular (Extended Data Fig. 2b), and we determined that tagging or click labelling does not compromise channel function (Extended Data Fig. 3a–e).

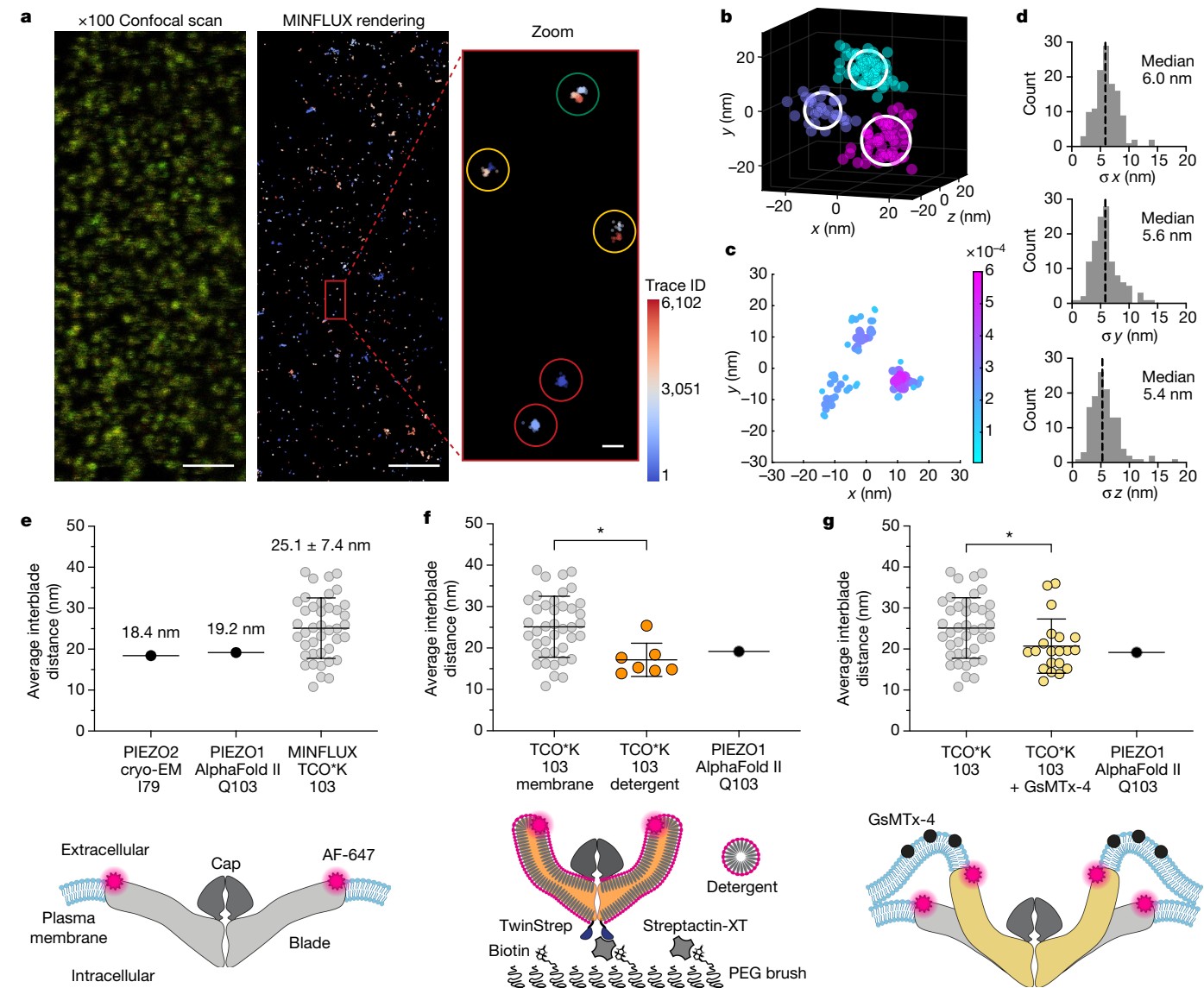

**Fig. 2 | The plasma membrane expands the blades of PIEZO1. a**, MINFLUX imaging and PIEZO1 trimer identification. Representative ×100 confocal scan (from *n* = 5 cells) of a HEK293 cell membrane expressing labelled TCO*K 103 PIEZO1 (scale bar, 2 µm) (left). Colour represents scaled intensity in each detector (650–685 nm (green) and 685–720 nm (red)). Clusters of raw localizations (traces) rendered as 15-nm spheres from the same region (scale bar, 2 µm) (middle). Zoomed-in rendering (3-nm spheres) showing single-labelled (red), double-labelled (yellow) and triple-labelled (green) PIEZO1 identified by the algorithm (scale bar, 20 nm) (right). **b**, Representative PIEZO1 trimer. Raw localizations (3-nm spheres) coloured by DBSCAN cluster. The centre positions were determined by a 3D GMM fit (white circles). **c**, Ostu-thresholded super-particle with threefold symmetry enforcement (*n* = 5 cells and *n* = 41 molecules). **d**, Histograms of GMM fit error for each trimer fluorophore position (*n* = 41 molecules and *n* = 123 fluorophore positions). Bin width = 1 nm.

The black dashed line indicates median fit error. **e**, Interblade distances of PIEZO1 in a membrane (grey circles; *n* = 41 molecules and *n* = 5 cells), the PIEZO2 structure (PDB ID: 6KG7) at the nearest aligned position and the AlphaFold II prediction (E2JF22) (top). A schematic of labelled PIEZO1 in a membrane is also shown (bottom). **f**, Interblade distances per detergent-solubilized PIEZO1 (17.1 ± 4.0 nm; orange circles; *n* = 7 molecules), AlphaFold II prediction (black circle) and PIEZO1 in a membrane (grey circles) (top). Kolmogorov–Smirnov test: *P = 0.0106 and *D* = 0.662. A schematic of the protein immobilization method and blade compaction is also shown (bottom). **g**, Interblade distances in a cell exposed to 20 µM GsMTx-4 (20.7 ± 6.6 nm; yellow circles; *n* = 20 molecules and *n* = 3 cells), AlphaFold II prediction (black circle) and PIEZO1 in a membrane (grey circles) (top). Kolmogorov–Smirnov test: *P = 0.0126 and *D* = 0.4341. A schematic of blade compaction from GsMTx-4 is also shown (bottom). All statistical tests are two-tailed. Values in **e**–**g** are mean ± s.d.

After labelling, cells were fixed in an isosmotic crosslinking solution to prevent changes in cell morphology and imaged without permeabilization to keep the plasma membrane intact. Collectively, our labelling system introduces a fluorophore into the amino acid chain of PIEZO1 with less than 1-nm physical offset error[25] and captures the channel in a native cellular environment.

To compare the resting conformation of PIEZO1 in a cell membrane to structural models, we first imaged labelled cells with 3D interferometric

photoactivation localization microscopy (iPALM)[26], a technique previously used to measure bulk conformational changes in membrane proteins[27]. Individual triple-labelled PIEZO1 particles localized to the plasma membrane were identified and segmented using a custom algorithm (Fig. 1b and Extended Data Fig. 4). Here clusters of three localization densities were identified from probability density renderings and segmented as individual particles (see Methods). As the effective localization errors for each molecule position are near the

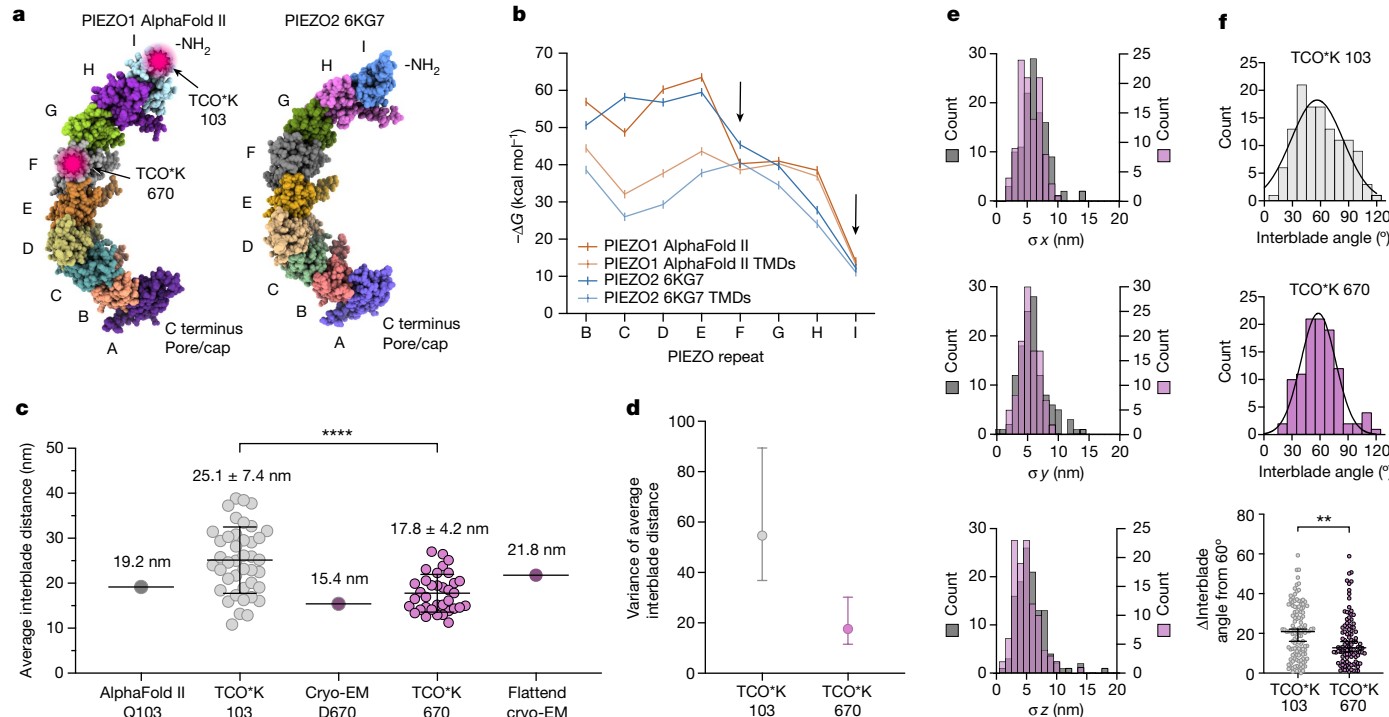

**Fig. 3 | Flexibility determines the extent of blade expansion. a**, PIEZO repeat domains of PIEZO1 (left; AlphaFold II E2JF22) and PIEZO2 (right; PDB ID: 6KG7), with tagged positions labelled. **b**, Inter-PIEZO repeat binding energy for PIEZO1 (blue; AlphaFold II E2JF22) and PIEZO2 (orange; PDB ID: 6KG7) with the contribution of transmembrane domains only (TMDs) or with additional contribution of ordered loops. Tagged repeats are denoted by black arrows. **c**, Scatter plot of average interblade distances at proximal blade position 670 (17.8 ± 4.2 nm; purple circles; $n = 35$ molecules and $n = 5$ cells) compared with distal position 103 (25.1 ± 7.4 nm; grey circles; $n = 41$ molecules and $n = 5$ cells) measured with MINFLUX in a cell membrane. Kolmogorov–Smirnov test: ****$P = 0.000087$ and $D = 0.5157$. Distances are shown as mean ± s.d. **d**, Variance and 95% confidence interval of average interblade distances at positions 103 and 670 from the scatter plot data in part **c**. $F$-test of equality of variances:

$P = 0.000535$ and $F$-statistic = 3.10. **e**, Histograms of GMM fit error for all identified triple-labelled PIEZO1 molecules at position 103 (grey; $n = 123$ fluorophore positions) and position 670 (purple; $n = 105$ fluorophore positions). Bin width = 1 nm. Difference in median GMM fit error $\Delta\sigma x = 0.66$ nm, $\Delta\sigma y = 0.59$ nm and $\Delta\sigma z = 0.69$ nm. **f**, Histograms of interblade angles for identified triple-labelled PIEZO1 molecules binned by 10 nm and fit with a Gaussian at position 103 (56.4 ± 27.8 nm; $R^2 = 0.84$) (top) and position 670 (58.2 ± 18.5 nm; $R^2 = 0.94$) (middle). Values are shown as mean ± s.d. A scatter plot of the change in interblade angles from symmetric (60°) for positions 103 and 670 is also shown (bottom). Mann–Whitney test: **$P = 0.0059$, $U = 5090$. Error bars are shown as median and 95% confidence interval. All statistical tests are two-tailed.

interblade distance predicted by structures and obfuscated by additional sample drift and vibration error, we were unable to resolve meaningful interblade distances from individual particles. To increase signal to noise and resolution, we instead used a template-free 3D particle fusion algorithm[28] to fuse localizations from all identified channels and generate a super-particle of triple-labelled PIEZO1 (Fig. 1c). With previous knowledge of subunit stoichiometry, we enforced threefold symmetry during the creation of the super-particle as the registration algorithm tends to match regions of dense localizations[28,29] (Extended Data Fig. 5a,b).

The individual localization clouds in the super-particle corresponding to each fluorophore position are not spherical. Localizations from a relatively rigid point source emitter imaged with near-isotropic resolution and fused with a method that accounts for anisotropic localization uncertainty should resolve as a sphere, as demonstrated with labelled subunits of the nuclear pore complex[28]. However, the PIEZO1 blade localization clouds were elongated approximately threefold (Fig. 1c,d). We hypothesized that this elongation was due to the superposition of conformational states, possibly driven by considerable flexibility of the distal blade. Relative to the averaged snapshot provided in structural models, this raises the interesting possibility that the blades of individual channels are not conformationally uniform. We next isolated the localization clouds associated with each of the three blades and calculated the average pairwise distance to all localizations in neighbouring

blades (Fig. 1d,e). Compared with the interblade distance calculated from the AlphaFold II structure prediction, the most probable distance measured from PIEZO1 in a cell is 6.2 ± 5.9 nm more expanded (mean ± s.d.) (Fig. 1e). As the structural model is membrane-free, these data also suggest that the plasma membrane exerts sufficient bending stress to expand the blades of PIEZO1 at rest.

Testing these two hypotheses more precisely required direct observation of the blade positions of single PIEZO1 molecules. To accomplish this, we used MINFLUX, a 3D super-resolution fluorescence microscopy method capable of true isotropic resolution with only 5- to 6-nm localization error[30–32]. Our system was equipped with a 3D sample stabilization system that actively locks the sample position with less than 2-nm error[32], minimizing uncertainty from drift and vibration. We prepared cells exactly as for iPALM, imaged the labelled cells with MINFLUX and identified triple-labelled particles with a separate automated 3D identification and clustering algorithm (Fig. 2a; see Methods). Raw localizations were first separated into clusters and then each cluster was individually fit with a 3D Gaussian mixture model (GMM) to determine each fluorophore centre position and positional uncertainty[33] (Fig. 2b). We designated PIEZO trimers by selecting clusters of three fluorophore positions spatially separated from any other detected fluorophore position by more than 100 nm. For all localization clusters, the median fluorophore localization error in $x$, $y$ and $z$ was 6.0, 5.6 and 5.4 nm, respectively ($n = 5$ cells and $n = 123$ fluorophore positions;

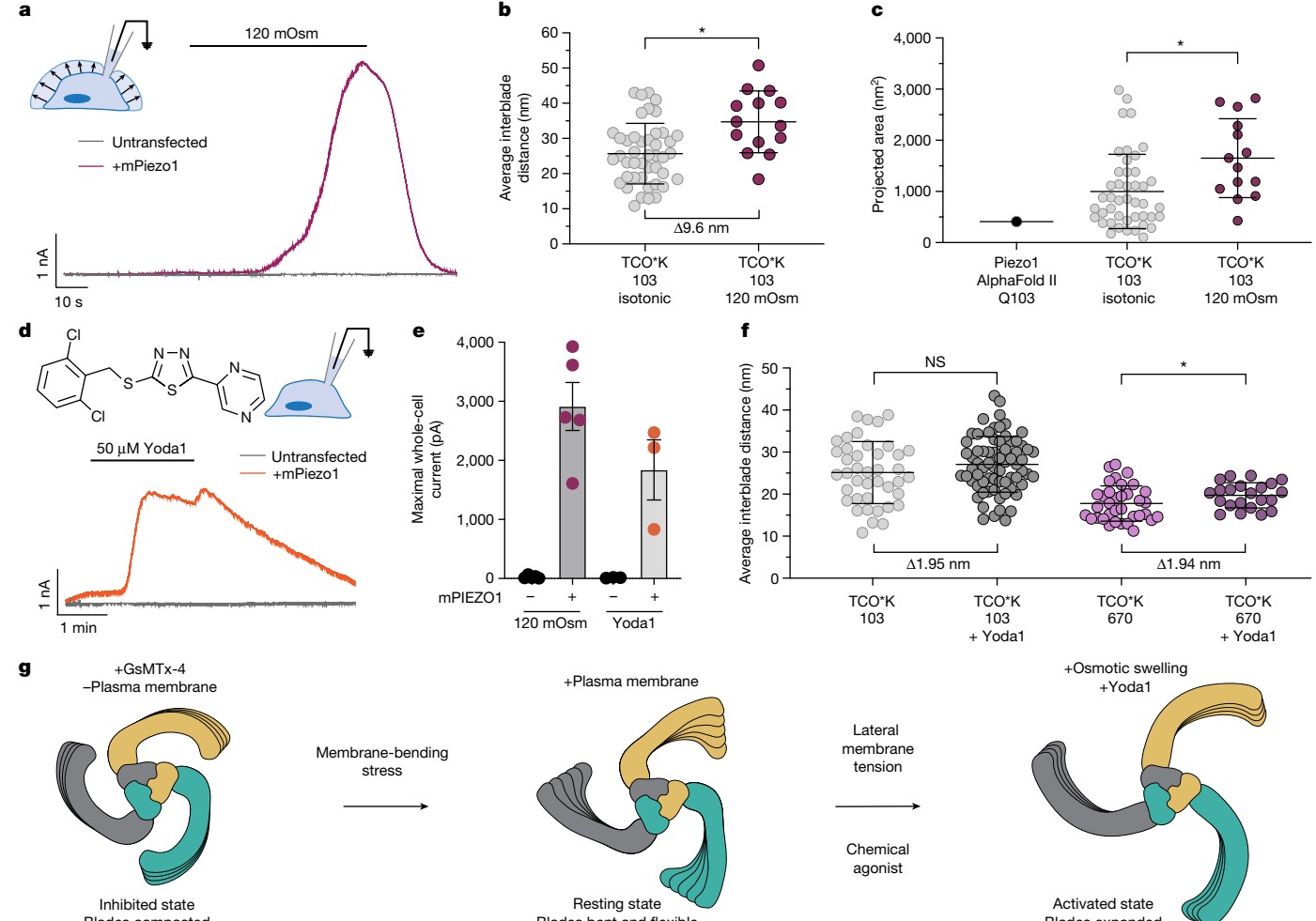

**Fig. 4 | Activation of PIEZO1 by blade expansion. a**, Representative whole-cell electrophysiology at +80 mV in response to hypotonic extracellular solution of *Swell1*-knockout HEK293F cells transfected with PIEZO1 and an untransfected control. **b**, Blade expansion at position 103 from osmotic swelling (34.7 ± 8.8 nm; red circles; *n* = 14 molecules and *n* = 4 cells) compared with an unstimulated cell (25.7 ± 8.6 nm; grey circles; *n* = 44 molecules and *n* = 5 cells). Kolmogorov–Smirnov test: *\*P* = 0.0169 and *D* = 0.474. **c**, Change in projected area calculated from circumradius using the distances in part **b** from osmotic swelling (1,651 ± 770 nm²; red circles), an unstimulated cell (999 ± 726 nm²; grey circles) and AlphaFold II prediction (411 nm²; black circle). Kolmogorov–Smirnov test: *\*P* = 0.0264 and *D* = 0.4513. **d**, Representative whole-cell electrophysiology at +80 mV in response to 50 μM Yoda1 of *Swell1*-knockout HEK293F cells transfected with PIEZO1 and an untransfected control. The chemical structure

of Yoda1 is also shown (top left). **e**, Maximal whole-cell currents from osmotic swelling and Yoda1 (*n* = 5 cells each (120 mOsm) and *n* = 3 cells each (Yoda1)). Values are shown as mean ± s.e.m. **f**, Interblade distances at position 103 with 50 μM Yoda1 (27.07 ± 6.60 nm; dark grey circles; *n* = 69 molecules and *n* = 3 cells) and the unstimulated condition (25.13 ± 7.39 nm; light grey circles; *n* = 41 molecules and *n* = 5 cells) (Kolmogorov–Smirnov test: *P* = 0.4712 and *D* = 0.1668), and at position 670 with 50 μM Yoda1 (19.71 ± 3.01 nm; dark purple circles; *n* = 22 molecules and *n* = 5 cells) and the unstimulated condition (17.77 ± 4.19 nm; light purple circles; *n* = 35 molecules and *n* = 5 cells) (Kolmogorov–Smirnov test: *\*P* = 0.0481 and *D* = 0.3714). Two significant figures are used to highlight precision. NS, not significant. **g**, Summary of results. Values in **b**,**c**,**f** are shown as mean ± s.d. All statistical tests are two-tailed.

Fig. 2d), far less than the calculated interblade distances (Extended Data Fig. 1b).

We created a fused super-particle from PIEZO1 TCO*K 103 centre positions obtained with MINFLUX and, like the iPALM super-particle, we again observed non-spherical elongation of the localization clouds (Fig. 2c). The minimal fluorophore position uncertainty also allowed us to directly measure distances from individual PIEZO channels. The average interblade distance between each blade at position 103 is 25.1 ± 7.4 nm (mean ± s.d.) (Fig. 2e), consistent with the most probable distance measured with iPALM (Fig. 1e). Although we expected the blades to expand to some degree given the observed PIEZO1–membrane dome flattening in large lipid vesicles[18,19], we were intrigued to find that, compared with the membrane-free AlphaFold II structural model, the distal regions of the blades of PIEZO1 are expanded on average by approximately 29% when embedded in a cell membrane. We also

note that the standard deviation of interblade distances is greater than the experimental localization error, supporting the hypothesis that the spread of distances is driven by intrinsic blade flexibility. Together, these data suggest that the blades of PIEZO1 are significantly expanded at rest, presumably by the plasma membrane, and that the distal regions of the blades are highly flexible.

## Extent of PIEZO1 blade expansion

We next tested whether the observed blade expansion in a cell is directly mediated by the plasma membrane. As the only existing solved structures of PIEZO1 lack the last approximately one third of the distal blade, we thus far have relied on structural models to calculate the relative extent of blade expansion. Thus, we directly measured the membrane-free channel conformation and compared it with the

AlphaFold II structural model. To do this, we expressed, solubilized and purified the PIEZO1 protein essentially as described for cryo-EM studies[4,5,7,17,21]. The distal blades of purified PIEZO1 were labelled at position 103 and immobilized onto a polyethylene glycol (PEG) brush surface for MINFLUX imaging in the presence of detergent (Fig. 2f, bottom). The channels were separated on average by more than 100 nm through sparse grafting of biotin-functionalized PEG, the minimum separation distance between channels required by the segmentation algorithm. We observed a significant decrease in interblade distance relative to the resting cellular state ($P = 0.0106$, Kolmogorov–Smirnov test) to $17.1 \pm 4.0$ nm (mean ± s.d.), very near the interblade distance measured from the structure prediction (Fig. 2f, top). These data confirm that the AlphaFold II structural model is a reasonable membrane-free comparison to our data. As removal of cellular components, including the membrane, compacts the blades, these data again suggest that the plasma membrane significantly expands the blades in a cell. These experiments also act as a critical control for our analysis pipeline in two key ways: our interblade measurements in detergent agree with existing structural models, and we observed a large conformational shift between each condition when the imaging and segmentation parameters are held constant.

The observed compaction of the blades upon removal of the plasma membrane and cellular components raises the possibility that inhibitors of channel activity can also act through the same mechanism. Some PIEZO1 inhibitors such as gadolinium, streptomycin and ruthenium red evidently block the flow of ions through the pore[1,34], but others, such as the gating modifier GsMTx-4, have no proven mechanism of action. GsMTx-4 is a peptide toxin isolated from the Chilean rose tarantula that broadly inhibits mechanosensitive ion channels[35,36]. The equilibrium binding constant $K_d$ of GsMTx-4 to a lipid bilayer and the half-maximal inhibitory concentration ($IC_{50}$) for PIEZO1 are both approximately 2 μM (refs. 36–38), consistent with the lipid bilayer being the primary target of action. In addition, molecular dynamics simulations suggest that GsMTx-4 acts as a mobile reserve of membrane material by shifting between shallow and deep penetration depending on bilayer tension, in effect acting as a buffer that reduces local membrane stress[37]. The net effect on PIEZO1 channel activity is a rightward shift of the current-displacement curve[36], requiring a much larger stimulus to open the channel. Given these models and the small size of the 35-amino acid toxin (approximately 2 nm in diameter; PDB ID: 1LU8), we reasoned that it may be able to directly embed into the membrane dome formed by PIEZO1 and release the local bending stress that keeps the blades extended (Fig 2g, bottom).

GsMTx-4 has a relatively low membrane affinity and a fast off-rate ($k_{off} \approx 0.2 \, \text{s}^{-1}$) (ref. 37). Thus, to best preserve the transiently inhibited conformational state, we applied GsMTx-4 for 5 min at a concentration (20 μM) ten times greater than the equilibrium binding concentration and quickly post-fixed the cells (see Methods). Using the same MINFLUX imaging method and segmentation algorithm as for previous experiments, we measured the interblade distance from single PIEZO1 channels. In the presence of GsMTx-4, the average interblade distance was $20.7 \pm 6.6$ nm (mean ± s.d.), which is significantly decreased relative to the cellular resting condition ($P = 0.0126$, Kolmogorov–Smirnov test), and very near the detergent-solubilized state and membrane-free structure prediction (Fig. 2g, top). These data indicate that release of membrane-bending stress causes blade compaction and provide a basic mechanism of inhibition for GsMTx-4. As GsMTx-4 appears to act specifically on the lipid bilayer, these data also suggest that blade expansion is mediated primarily through the plasma membrane rather than through tethering to the cytoskeleton or extracellular matrix[39,40].

## Analysis of PIEZO1 blade flexibility

Next, we focused on to what extent the apparent spread of conformational states is driven by blade flexibility. Although the observed heterogeneity in blade expansion might be due in part to local differences in membrane properties, such as topography and bending modulus, the long blade of transmembrane domains should roughly behave like a flexible elastic rod within the confines of the plasma membrane. Deflections from random thermal energy at the distal ends should be larger than near the centre of the channel. Indeed, physical models predict that at least part of PIEZO1 might be similarly flexible as a lipid bilayer, implying that thermal fluctuations alone can result in substantial deformations of the shape of PIEZO1 (refs. 18,19). However, given the large distribution of conformational states, we wondered whether certain structural features might be responsible for these mechanical properties.

The stiffness of the tertiary structure of a protein is predominantly determined by the strength of amino acid interactions at binding interfaces[41,42]. Each blade of PIEZO1 can be divided into nine PIEZO repeat domains, with each repeat forming a cluster of four packed transmembrane helices containing inter-repeat binding interfaces (Fig. 3a). Using the AlphaFold II structure of the PIEZO1 blade, we calculated the inter-PIEZO repeat binding energy ($-\Delta G$) for domain interfaces[43], with and without the contribution of intracellular and extracellular loops, including domains expected to increase interdomain binding strength, such as the beam (Fig. 3b). We observed a dramatic, graded decrease in $-\Delta G$ along the proximal to distal axis of the blade, consistent with PIEZO2 repeat domain binding energies calculated directly from the cryo-EM structure. Low binding energy is especially apparent for distal repeat I, which binds to only one other PIEZO repeat and is exposed more extensively to the plasma membrane. The average free energy difference between repeat F—at the edge of the resolved PIEZO1 cryo-EM structures—and repeat I—the most distal repeat—is 28.6 kcal mol$^{-1}$ (16.9 $k_B T$). In a complex membrane-water environment, actual interface binding energies are probably different, but the general trend indicates that proximal repeats close to the pore domain are more rigid than distal repeats and less susceptible to bending by the plasma membrane.

To measure flexibility at more proximal blade domains within resolved cryo-EM structures, we tagged PIEZO1 with TCO*K at amino acid 670, which lies in an extracellular loop of repeat F (Extended Data Figs. 1b and 2b). When measured and analysed with our MINFLUX pipeline, we observed a statistically significant decrease in interblade distances relative to repeat I ($P = 0.000087$, Kolmogorov–Smirnov test) (Fig. 3c). Of note, the variance of average interblade distances was significantly decreased between positions 103 and 670 (Fig. 3d), indicating that a smaller range of conformational states are being occupied at repeat F and that there is a large difference in flexibility. GMM fit errors for each condition are nearly the same (change in median GMM fit error ($\Delta \sigma$) $x = 0.66$ nm, $\Delta \sigma y = 0.59$ nm and $\Delta \sigma z = 0.69$ nm; Fig. 3e), so are not responsible for the apparent difference in mechanical properties. Consistent with decreased flexibility, the angle between the three fluorophore positions on each identified PIEZO1 molecule at repeat F was significantly more symmetrical than at repeat I (Fig. 3f). We suspect that such differences in flexibility are why the distal blade has not yet been resolved by cryo-EM.

We next examined the relative extent to which each section of the blade is expanded by the plasma membrane. Compared with the membrane-free structures and predictions, the blade at proximal repeat F is expanded by only $2.4 \pm 4.2$ nm, compared with $5.9 \pm 7.4$ nm at distal repeat I (mean ± s.d.) (Fig. 3c). This more than twofold average increase is consistent with the vast difference in energetic stability of these domains and the consequent difference in flexibility. The average interblade distance of position 670 in a cell lies in between that of the presumably stress-free detergent-solubilized structure and the highly strained, flattened cryo-EM structure of PIEZO1 solved outside-out in 10-nm lipid vesicles[21], providing additional evidence that we have captured a resting state of blade expansion by the plasma membrane.

We next asked whether we could measure induced changes in blade flexibility at a single tagged position. To do this, we altered membrane stiffness by changing the lipid composition of the plasma membrane. Saturated fatty acids, such as margaric acid, increase membrane stiffness and viscosity[44] and should consequently decrease the magnitude of blade displacement by random thermal motion. We enriched the plasma membrane of cells expressing PIEZO1 with 300 μM margaric acid, imaged with MINFLUX, and found that the distal blades had significantly decreased variation of interblade distances, consistent with an apparent decrease in the magnitude of blade fluctuations (Extended Data Fig. 6b). These data further support the observation that the spread of conformational states at the distal blade is driven by intrinsic blade flexibility. We observed no significant change in interblade distance with margaric acid enrichment (Extended Data Fig. 6a). Mathematical modelling of PIEZO1 in the plasma membrane predicts that increased membrane stiffness should also expand the blades and decrease the apparent channel gating threshold[4]; however, electrophysiological data conversely indicate that margaric acid increases the gating threshold[45]. These data suggest that the influence of membrane composition on the conformation of PIEZO1 is probably more nuanced than predicted from modelling alone and may involve direct protein binding and modulation.

## Activation of PIEZO1 by blade expansion

GsMTx-4 inhibits PIEZO channel activity, and our data suggest that it compacts the blades of PIEZO1 by releasing membrane-bending stress. Conversely, application of force to a cell membrane activates PIEZO1 (refs. 1,2), presumably through lateral membrane tension and deformation of the membrane dome[4]. Thus, we focused on to what extent the blades expand when the plasma membrane is stretched. We sought a stimulus compatible with MINFLUX imaging that uniformly expands the membrane and directly activates PIEZO1.

A hypotonic extracellular environment increases cell volume via osmotic swelling. Given a finite surface area, swelling applies tension to the plasma membrane[46]. Osmotic swelling also induces $Ca^{2+}$ influx through PIEZO1 (ref. 47), but we found no electrophysiological evidence that it directly activates the channel under normal conditions. Mechanically evoked PIEZO1 currents inactivate quickly ($\tau_{inactivation} = 10–30$ ms) at negative membrane potentials[1], whereas the rate of osmotic swelling in HEK293 cells is slow, reaching peak volume in approximately 2.5 min (ref. 48). We therefore suspected that fast inactivation obscures channel activation in response to osmotic swelling. Osmotic swelling also elicits an outward chloride current through the ubiquitous volume-regulated anion channel SWELL1 (refs. 49,50), further masking the PIEZO1-evoked currents. We circumvented both of these issues by measuring osmotically induced PIEZO1 activation in *Swell1*-knockout HEK293 cells with whole-cell voltage clamp at +80 mV at which $\tau_{inactivation}$ is approximately ten times slower than the negative-holding potentials at which PIEZO currents are typically recorded to simulate physiological conditions[51]. We observed large PIEZO1-dependent currents that tracked the time course of cell swelling (Fig. 4a), suggesting that membrane stretch from osmotic swelling can directly activate PIEZO1.

We next exposed cells expressing fluorescently labelled TCO*K 103 PIEZO1 to hypotonic solution for 2.5 min and immediately fixed the cells in a hypotonic fixative at the peak of cell swelling to preserve the activated state. When measured with MINFLUX, average interblade distances were significantly increased from 25.7 ± 8.6 nm in the resting state to 34.7 ± 8.8 nm in the swelled state (mean ± s.d.; $P = 0.0169$, Kolmogorov–Smirnov test), nearly twice as far on average as the extent of resting blade expansion from membrane-bending stress alone (Fig. 4b). This expansion corresponded with a significant increase in the total projected area of the channel relative to both the membrane-free structural models and the state of resting expansion (Fig. 4c), indicating that the PIEZO1 dome is flattening in response to this type of membrane stretch.

These results demonstrate that membrane stretch from osmotic swelling sufficient to activate the channel also stretches the blades of PIEZO1.

We also tested the small-molecule agonist Yoda1, which, like osmotically induced cell swelling, causes robust PIEZO1-dependent $Ca^{2+}$ entry into cells[52]. Yoda1 slows the rate of channel inactivation (Extended Data Fig. 3d,e) and significantly increases channel open probability in the absence of applied force[52]. The exact mechanism by which Yoda1 agonizes PIEZO1 is unknown, but molecular dynamics simulations and mutagenesis suggest that it binds in a pocket between PIEZO1 repeats A and B[53]. To compare the relative extent of channel activation with osmotic swelling, we measured whole-cell currents evoked by bath-applied 50 μM Yoda1 to *Swell1*-knockout HEK293 cells held at a positive membrane potential (Fig. 4d). We also observed large (more than 1 nA) currents from bath-perfused Yoda1 without mechanical stimulation, suggesting that these two distinct stimuli both robustly activate PIEZO1 (Fig. 4e).

We next measured Yoda1-induced changes in PIEZO1 interblade distances at position 103 with MINFLUX. When cells were incubated with and fixed in the presence of 50 μM Yoda1, the average interblade distance increased on average by 1.95 nm ($P = 0.4712$, Kolmogorov–Smirnov test; Fig. 4f). Given the wide spread of conformational states from domain flexibility, we did not observe a statistically significant change in distance at this position. Therefore, we also measured Yoda1-induced blade movement at position 670, a more rigid location within the blade that occupies a smaller range of conformational states (Fig. 3). Here we observed a statistically significant change in average interblade distance of 1.94 nm on average ($P = 0.0481$, Kolmogorov–Smirnov test), which is remarkably close on average compared with the Yoda1-induced change in distance at position 103 (Fig. 4f). These data indicate that Yoda1 directly expands the blades, inducing a small, stereotyped conformational movement upon binding. These data also highlight the power of our technique to accurately observe nanometre-scale molecular movements.

## Discussion

In this study, we have shown how the cellular environment can shape the conformation of PIEZO1 using direct nanoscopic fluorescence imaging. Relative to published structures, the blades of PIEZO1 are significantly expanded by the plasma membrane, consistent with quantitative predictions of the elastic properties of the PIEZO–membrane dome[4,8,18,19]. We have shown that the blades of PIEZO1 are highly flexible, which probably has important implications for the properties of force transmission from the membrane to the pore domain and may explain why the distal domains of PIEZO1 have not been resolved by cryo-EM. We have also shown how blade expansion by chemical and mechanical modulators corresponds with channel activation. Together, these data provide a foundation for understanding how PIEZO1 is activated in a cellular environment (Fig. 4g).

In experiments designed to measure PIEZO1 activity in the nominal absence of tension, PIEZO1 is spontaneously active in a cell membrane (resting open probability $P_{open} \approx 0.5\%$)[54,55], and application of GsMTx-4 appears to inhibit this spontaneous activity[36]. We have shown that both application of GsMTx-4 and removal of the membrane with detergent contract the blades of PIEZO1 relative to the cellular resting state. These data suggest that membrane-free structural models represent a state in which $P_{open}$ is presumably zero and the overall conformation is in its lowest energy shape. We have also shown that the plasma membrane directly acts to expand the blades of PIEZO1 at rest, in agreement with previous physical models[4,18,19]. The extent of this membrane-mediated expansion is exceptionally large, a property conferred by high blade flexibility. An important consequence of low rigidity is that less external force is required to flatten and gate the channel. This not only confers a high sensitivity to membrane tension[4,18,19] but might also allow PIEZO1 to more frequently sample an open state without external force. Like

the resting tension from tip links exerted on the inner-ear hair cell transduction channel complex[56–59], a large resting blade expansion enabled by flexible blade domains may be a basic mechanism to maintain the channel in a responsive state and confer a specific range of tension sensitivity.

We suggest that the blades may not be uniformly flexible, a property which could be imparted by variable binding strength between domains (Fig. 3b). These calculations have important implications for the overall structure and function of the membrane dome, the collective structure of the plasma membrane and the PIEZO protein[4]. Graded changes in compliance towards the distal ends of the blades may enable a smooth mechanical transition between the rigid centre of the channel and the relatively flexible plasma membrane. The more compliant distal edge of the PIEZO1 dome may also allow the channel to dampen low-magnitude mechanical noise in a cell. The distal portions of the blades appear to move relatively independently of each other within a single channel complex, even in the presence of membrane tension (Extended Data Fig. 7). These data suggest that the distal blades do not move cooperatively during gating. Future studies might test whether and how graded blade flexibility impacts protein function by altering inter-PIEZO domain binding strength with point mutagenesis or double-cysteine crosslinking, the latter possibly allowing for acute and reversible manipulation of mechanical properties.

The apparent correlation between the extent of blade expansion and channel activity demonstrates the importance of the local membrane environment in determining channel properties. For example, alteration of membrane lipid composition can modulate the gating and inactivation properties of PIEZO1 (ref. 45), and we have demonstrated that increasing membrane stiffness with the saturated fatty acid margaric acid can alter the mechanical properties of blades (Extended Data Fig. 6). The lipid composition of membrane microdomains or local membrane topography might directly modulate resting channel open probability via blade conformational changes and, consequently, alter the amount of force required to open the pore. Such features could tune the mechanical response properties of the channel and may be present in specialized sites of mechanotransduction, such as Merkel cell–neurite complexes[11,60]. For example, ultrastructural features, such as the filamentous connections between hair follicle epithelial cells and low-threshold mechanoreceptor lanceolate endings[61], may serve to modulate the amount of bending stress applied to the blades of PIEZO2 proteins, perhaps altering channel activity to link the particular anatomical properties of mechanoreceptors to their distinctive functional outputs.

The data described here show that conformational dynamics of individual membrane proteins have been observed with direct fluorescence nanoscopy at the level of single molecules. Although methods such as single-molecule FRET can resolve relative fluorophore positions in membrane proteins with nanometre accuracy within the spatial distance required for resonance energy transfer[62–64], our approach reports absolute positions without a constrained radius of action. We expect that these methods will provide a foundation for the use of fluorescence nanoscopy for single-molecule structural biology, especially for proteins with highly flexible domains or for those refractory to study with current electron microscopy methods. Increased effective labelling efficiency with methods such as DNA-PAINT[65], increased signal to noise with microscopy methods such as MINSTED[66] and increased ability to process complex imaging datasets with more advanced computational methods will probably advance our ability to resolve the structure of proteins embedded within the complex milieu of a cell.

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

## Methods

No statistical methods were used to predetermine sample size. The experiments were not randomized and investigators were not blinded to allocation during experiments and outcome assessment.

### Expression constructs

The coding sequence of mouse PIEZO1 (E2JF22, UniprotKB entry) was codon-optimized, synthesized and cloned into the pcDNA3.1 plasmid. An amber stop codon (TAG) was inserted at the noted amino acid positions via site-directed mutagenesis with the Q5 Site-Directed Mutagenesis Kit (New England Biolabs). The coding sequence for HaloTag was amplified from the pHTC HaloTag CMV-neo vector (Promega) and the coding sequence for mEos3.2 was amplified from the mEos3.2-ER-5 vector (Addgene). The coding sequence for Strep-Tag II was codon-optimized and synthesized (IDT). The HaloTag and mEos3.2 sequences were separately subcloned along with the Strep-Tag II into the *mPiezo1*-pcDNA3.1 plasmid with the NEBuilder HiFi DNA Assembly kit (New England Biolabs). The sequence of each plasmid was verified by whole-plasmid sequencing before use. All DNA sequences were viewed and designed in SnapGene software (Dotmatics).

### Cell preparation for imaging

Cells were prepared and labelled the same for both iPALM and MINFLUX imaging (Extended Data Fig. 2). HEK293F cells (Expi293, Thermo Fisher) were grown in Expi293 expression medium (Thermo Fisher) to a density of $1–2 \times 10^6$ cells per ml. At all times, the cells were maintained at 37 °C with 8% $CO_2$ shaking at 125 rpm on a rotator with a 19-mm orbit diameter. Cells were verified to be free of mycoplasma using the using the MycoAlert Mycoplasma Detection Kit (Lonza). Before transfection, the cells were centrifuged at $100g$ for 3 min, exchanged into fresh medium containing 250–500 µM of *trans*-cyclooct-2-en-L-lysine (axial isomer) (SiChem), and transferred to a culture flask. Each flask was transfected with a 1:1 ratio of pNEU-hMbPylRS-4xU6M15 and a PIEZO1 expression construct at a total concentration of 2 µg ml$^{-1}$ using either 40 kDa PEI (PolySciences) or EndoFectin Expi293 transfection reagent (GeneCopeia). The transfected cells were allowed to express for 24–36 h. The cells were then twice iteratively pelleted and resuspended in fresh Expi293 medium and cultured for 30 min to let excess TCO*K diffuse from the cells.

To label the cells, $1.5 \times 10^6$ cells were moved to a 1.5-ml Eppendorf tube and the volume was brought up to 1 ml with fresh Expi293 medium containing a final concentration of 1% w/v blocking reagent, either BSA (Sigma) or Roche Blocking Reagent (Roche). The cells were mixed gently and allowed to block for 3 min at room temperature, centrifuged at $100g$ for 2 min and resuspended in Expi293 medium + 1% w/v blocking reagent + 4 µM tetrazine–Alexa Fluor 647. The cells were incubated for 10 min at room temperature away from light with occasional end-over-end mixing. To removed excess fluorophore, the cells were pelleted and washed three times in Expi293 medium + 1% blocking reagent, and then once in Expi293 medium without blocking reagent. During the washing steps, great care was taken to be as gentle as possible. The cells were finally diluted to a concentration of $0.3 \times 10^6$ cells per ml in Expi293 medium and plated directly onto coverslips.

### Coverslip preparation for iPALM imaging

Circular coverslips 25 mm in diameter with embedded wide spectral band gold fiducials ($600 \pm 100$ nm) under a 50-nm $SiO_2$ layer (Hesztig) were first prepared by washing with 100% ethanol and drying with a stream of purified air. The surfaces of the coverslips were next rendered hydrophilic by incubation with 1 M KOH for 5 min. The coverslips were washed in MilliQ water, and again dried with a stream of purified air. Of labelled cells at $0.3 \times 10^6$ cells per ml in Expi293 medium, 400 µl was plated onto the coverslips and allowed to adhere to the glass surface at 37 °C in a cell culture incubator for 15 min.

After the cells were adhered, they were washed with pre-warmed 37 °C 1× Hank's balanced salt solution (HBSS) without $Ca^{2+}$ or $Mg^{2+}$ and fixed in pre-warmed 37 °C 1× HBSS containing 0.8% paraformaldehyde (PFA) and 0.1% glutaraldehyde for 10 min. Special care was taken to gently pipette the solutions at this stage, so as not to mechanically disturb the cells. The cells were washed and quenched in 1× HBSS containing 50 mM Tris (pH 7.4) for 5 min and then washed extensively in 1× HBSS.

The coverslip was exchanged into an isotonic imaging buffer (50 mM Tris-HCl (pH 8.0), 10 mM NaCl, 3.33% glucose, 100 mM cysteamine, 40 µg ml$^{-1}$ bovine-liver catalase and 100 µg ml$^{-1}$ glucose oxidase from *Aspergillus niger*, type VII) by successive washing, and then overlaid with a plain 25-mm KOH-treated coverslip and sealed using 5-min epoxy (ITW Performance Polymers) and Vaseline (Unilever). The coverslip was mounted onto the microscope as previously described[26]. Cells were prepared and imaged on the same day.

### iPALM data acquisition

In brief, 1–2 cells were isolated in the imaging field of the iPALM microscope, and the instrument was calibrated using embedded gold fiducials as previously described[26]. Imaging was performed with custom LabView software as previously described[26]. To capture blinking of Alexa Fluor 647, samples were imaged with 30-ms exposure and 3 kW cm$^{-2}$ 640-nm laser excitation for 60,000 frames captured using three EMCCD cameras (iXon 897, Andor). Although not used in downstream architecture analysis due to effective labelling inefficiency and image registration error, C-terminally tagged mEos3.2 was also imaged with 561-nm laser excitation and 405-nm laser activation for 20,000–120,000 frames.

### iPALM data pre-processing

Image reconstruction was performed using PeakSelector (G. Shtengel and H. Hess, Howard Hughes Medical Institute, https://github.com/gleb-shtengel/PeakSelector) as previously described[26]. Gold nanoparticles embedded in the coverslip were used as fiducial markers to calibrate, align and transform overlaid frames into a single 3D image. Localizations with estimated $x/y$ uncertainty of more than 0.06 pixels (or nanometre equivalent) were filtered out of the data. Only localizations that were less than 150 nm of the coverslip fiducials were included to isolate those found at or near the plasma membrane. Localizations were exported from PeakSelector as an ASCII file, and total raw localization data were exported as a TIFF file. Custom MATLAB software was used to remove fiducial bead localizations by identifying beads in the total raw data image and eliminating corresponding bead localizations in the ASCII file (Extended Data Fig. 4b).

### PIEZO molecule segmentation and 3D particle fusion

A summed $Z$-projection of pre-processed localizations were rendered at 3 nm per pixel in PeakSelector using standard settings and saved as a TIFF file. The rendered image and an ASCII file containing pre-processed localizations were loaded into MATLAB and candidate triple-labelled PIEZO1 molecules were identified and segmented. First, peaks were found in the rendered image by first bandpass filtering the data and using Crocker and Grier's algorithm[67] to identify peaks. Peaks were next subjected to nearest neighbour analysis, requiring that each fluorophore position must have two neighbours, that their centre positions separated by more than 9 and less than 60 nm, and that each peak in the cluster be greater than 60 nm away from any other localizations. Each rendered candidate PIEZO1 molecule was then connected to the corresponding localizations, and the localizations were segmented into individual particles (Extended Data Fig. 4c). By using a summed $Z$-projection, this approach is limited to segmentation in the $x–y$ plane. To remove unassociated localizations in $z$, any localizations more than 75 nm away from the particle mean were removed from each particle.

Each segmented particle was next fed into the template-free single-particle averaging workflow by Heydarian et al.[28]. In brief, the scale-sweep approach was used to determine an optimal scale parameter of 5 nm for the all-to-all registration process between segmented particles. A threshold of 1 was used for five iterations of the Lie algebra consistency check, which was used to form a data-driven template and create an initial set of aligned particles. An additional processing step was added to rotate these initial aligned particles into the $x$–$y$ plane before promoting a threefold symmetry by rotating each initially aligned particle by a random integer factor of $2 \times \pi/3$. These symmetry-promoted particles were then used to create a final super-particle in a bootstrapping step, which compared each particle to the data-driven template.

Super-particles (Figs. 1c,d and 2c and Extended Data Fig. 5) were visualized by calculating a kernel density estimate for the final super-particle, using the MATLAB function mvksdensity and a bandwidth of 2.5 nm. Localizations in the particle were plotted with size and colour proportional to their local density.

## Sample preparation for MINFLUX imaging

Number 1.5 round glass coverslips (Warner Instruments) of 18-mm diameter were cleaned by boiling in 1% Hellmanex III detergent (Hellma GmbH) in MilliQ water and sonicating for 10 min in a water bath. The coverslips were washed five times in MilliQ water, sonicated in 1 M KOH for 10 min and then again washed in MilliQ water. The coverslips were exchanged into 100% ethanol and stored covered at room temperature for up to 1 week.

Before plating, coverslips were dried with a stream of purified air. Of the $0.3 \times 10^6$ cells per ml cell suspension in Expi293 medium, 207 µl was plated onto the coverslips and allowed to adhere to the glass surface at 37 °C in a cell culture incubator for 15 min. The cells were washed and fixed as for iPALM imaging.

For GsMTx-4 experiments, the plated cells were washed in 1× HBSS and then incubated with 20 µM GsMTx-4 (Abcam) for 5 min at room temperature. The solution was removed completely and fixative (pre-warmed 37 °C 1× HBSS containing 0.8% PFA and 0.1% glutaraldehyde) was immediately but gently added. The cells were allowed to fix for 10 min and washed in HBSS.

For margaric acid enrichment of the plasma membrane, enrichment was performed essentially as previously described[45]. In brief, a fresh ampule of margaric acid (Nu-Chek Prep) was dissolved to 150 mM in DMSO. Margaric acid stock was added to warmed Expi293 medium at a final concentration of 300 µM. The medium was alternately vortexed, sonicated and incubated at 37 °C until completely dissolved. Six hours after transfection, the medium was exchanged for the margaric acid-enriched medium. The cells were cultured for an additional 18 h before preparation for imaging.

For Yoda1 experiments, a stock solution of 10 mM Yoda1 (Tocris) in DMSO was added to pre-warmed 37 °C 1× HBSS to a final concentration of 50 µM. The solution was vortexed at full speed for 45 s to completely dissolve the Yoda1. The cells were washed in 1× HBSS and the Yoda1 immediately added. The cells were incubated for 5 min at room temperature. Next, the solution was removed and fixative (pre-warmed 37 °C 1× HBSS containing 0.8% PFA and 0.1% glutaraldehyde) containing 50 µM Yoda1 was immediately but gently added. The cells were allowed to fix for 10 min and washed in HBSS.

For osmotic swelling experiments, the plated cells were washed in 1× HBSS and then exposed to a 120 mOsm modified Ringer's solution (48.8 mM NaCl, 5 mM KCl, 10 mM HEPES (pH 7.40) and 10 mM D-glucose) for 2.5 min at room temperature. Next, the cells were gently exchanged into a hypotonic fixative (120 mOsm modified Ringer's, 0.8% PFA and 0.1% glutaraldehyde). The cells were allowed to fix for 10 min and washed extensively in 120 mOsm modified Ringer's solution. The osmolality of all solutions was determined to be ±5 mOsm with a vapour pressure osmometer.

After fixation and washing, 150-nm gold nanosphere fiducials (BBI Solutions) were applied to the coverslip and incubated for 5 min at room temperature. The coverslip was then washed in HBSS.

For imaging, the cells were exchanged into the isotonic imaging buffer as for iPALM, except with 20 mM cysteamine. The coverslip was placed onto a glass slide containing a cavity well (Globe Scientific) filled with imaging buffer and pressed down to remove excess buffer. The coverslip was then sealed onto the slide using Elite Double 22 dental epoxy (Zhermack).

## Protein expression, solubilization and immobilization for in vitro imaging

To obtain detergent-solubilized mouse PIEZO1 protein, Expi293 cells were transfected with mPIEZO1-N-tandem HisTag-TAG1 03-C-HaloTag-TwinStrep with pNEU-hMbPylRS-4×U6M15 in a 30 ml culture containing 500 µM TCO*K. After 12–16 h, cells were fed with 7 ml Expi293 medium and sodium butyrate was added to a final concentration of 5 mM. After 48 h, the cells were pelleted and resuspended in fresh Expi293 medium and cultured for an additional hour to let excess TCO*K diffuse from the cells. The cells were then washed by pelleting at 100$g$ and resuspending twice in ice-cold 1× HBSS containing 1× HALT protease inhibitor (Thermo Fisher). The cells were pelleted for a final time at 1,000$g$, the supernatant removed and the cell pellet was flash frozen in liquid nitrogen and stored at −80 °C.

To affinity purify the protein, frozen cell pellets were directly resuspended in ice-cold solubilization buffer (25 mM HEPES, 150 mM NaCl, 2 mM DTT, 1% C12E9 and 1× HALT protease inhibitor). The mixture was rotated end-over-end at 4 °C for 1 h to solubilize membrane proteins and centrifuged at 45,000$g$ for 30 min at 4 °C to pellet non-soluble debris and aggregates. The supernatant was loaded onto a column containing 1 ml settled TALON metal affinity resin pre-washed with wash buffer (25 mM HEPES, 150 mM NaCl, 2 mM DTT, 0.1% C12E9 and 1× HALT protease inhibitor) and the His-tagged PIEZO1 protein was allowed to bind. After washing the resin with 30 ml of wash buffer, the column was capped and 300 µl wash buffer containing 4 µM tetrazine–Alexa Fluor 647 was added. The resin bed was resuspended and incubated for 10 min at room temperature, away from light. The resin was washed extensively in wash buffer without protease inhibitor. The protein was eluted in 25 mM HEPES, 150 mM NaCl, 2 mM DTT, 0.1% C12E9 and 200 mM imidazole. The eluate was then directly loaded onto a column containing 1 ml Streptactin Sepharose resin pre-washed with wash buffer. After binding, the resin was washed with 30 ml wash buffer and eluted in wash buffer containing 25 mM biotin. The eluate was concentrated on an Amicon 50-kDa molecular weight cut-off column at 5,000$g$ and washed twice with wash buffer. Finally, the protein was buffer exchanged using two 40-kDa molecular weight cut-off Zeba desalting columns pre-equilibrated with wash buffer. The protein concentration was quantified using A280 on a Nanodrop (Thermo Fisher Scientific), split into aliquots, flash frozen on liquid nitrogen and stored at −80 °C.

Number 1.5 glass coverslips of 22 × 22 mm pre-functionalized with a sparsely biotinylated PEG brush were purchased (Microsurfaces). Given the coating density, the average distance between biotins on the surface of the brush is approximately 112 nm, which is approximately the same cut-off distance used by the clustering algorithm to identify triple-labelled PIEZOs. All steps were performed at room temperature. First, the coverslips were incubated with undiluted 150-nm gold nanosphere fiducials (BBI solutions) for 20 min. These gold nanospheres bound sparsely into imperfections in the PEG brush surface, but densely enough such that at least two gold fiducials could be found in a field of view for stabilization on the MINFLUX microscope. The coverslips were then washed well with immobilization buffer (25 mM HEPES (pH 8.0), 150 mM NaCl, 2 mM DTT, 0.1% C12E9 and 1% Roche blocking reagent) and incubated in this buffer for 15 min to block any unpassivated sites. Next, non-functionalized Streptactin-XT (IBA Lifesciences)

was diluted to 100 nM in immobilization buffer, added to the coverslip for 7 min at room temperature to adhere to the biotins on the PEG brush, and the coverslips were washed well with immobilization buffer to remove excess Streptactin-XT. Flash-frozen protein was thawed on ice, diluted to 20 nM in immobilization buffer and applied to the coverslip. Strep-tagged PIEZO1 was allowed to bind for 10 min to the immobilized Streptactin-XT and the coverslip was again washed thoroughly in immobilization buffer. The protein was then exchanged into and washed with immobilization buffer containing the detergent GDN without Roche blocking reagent (25 mM HEPES (pH 8.0), 150 mM NaCl, 2 mM DTT and 0.02% GDN).

The coated coverslip was placed onto a glass slide containing a cavity well (Globe Scientific) filled with imaging buffer (50 mM Tris-HCl (pH 8.0), 150 mM NaCl, 10% glucose, 0.02% GDN, 20 mM cysteamine, 40 µg ml$^{-1}$ catalase from bovine liver and 100 µg ml$^{-1}$ glucose oxidase from *A. niger*, type VII) and pressed down to remove excess buffer. The coverslip was then sealed onto the slide using Elite Double 22 dental epoxy (Zhermack) and mounted onto the MINFLUX microscope. Imaging was performed as described above.

## MINFLUX data acquisition
All MINFLUX data were acquired on a commercial MINFLUX 3D microscope using Imspector software with MINFLUX drivers (Abberior Instruments). A field of view was chosen with three or more gold fiducials for stabilization. An active stabilization system that uses near-infrared scattering from gold fiducials and active-feedback correction was used to lock onto a chosen spatial set point. It was ensured that the mean standard deviation of the measured sample position relative to the stabilization set point set by the MINFLUX interface software was less than 3 nm in each axis[32]. A 25–225 µm$^2$ field of view at the bottom of the cell was chosen for MINFLUX imaging. More than 50% of the visible fluorophores within the field of view were driven into a dark state using iterative confocal scans with the 640-nm laser at 2–4% power. The sample was imaged with 6% 640-nm laser power, manually ramped up to 9% over the course of the imaging session. Then, 405-nm laser power was slowly ramped up from 0 to 12–18% over the course of several hours. Samples were imaged for 12–48 h total. At least three separate biological and experimental replicates were imaged for each condition. The 640-nm excitation laser was measured to be approximately 4.30 µW per percent set power at the sample plane, and a 405-nm activation laser was measured to be approximately 16 nW per percent set power at the sample plane. Note that during the MINFLUX targeting routine, the laser power is ramped up to a final factor of six in the last iteration[32].

## MINFLUX data analysis
Raw final valid localizations from the last targeting iterations were exported directly from the MINFLUX Imspector interface as a .mat file. Custom MATLAB analysis software was then used to identify and segregate clusters of three localizations. To be as consistent and unbiased as possible, all data were analysed with the same parameters, except for one special case (see below). First, the data were filtered to remove traces with a standard deviation of more than 10 nm and containing more than three localizations per trace. This step removed localizations from background and large streaks, which were probably due to diffusing fluorescent molecules moving through the imaging plane. Next, the localizations were subjected to a density-based clustering algorithm essentially as previously described[33]. This algorithm uses two-step DBSCAN clustering (dbscan2 in MATLAB) followed by an expectation maximization GMM to assign 3D localizations to the position of fluorophores. Here the first DBSCAN step had an epsilon of 30 nm and required five neighbours for a core point (minpts = 5). The second DBSCAN step had an epsilon of 6–7 nm, depending on the amount of noise in the data, and minpts = 5. The initial GMM fit sigma was set to 5 nm. The fluorophore centre positions were estimated as the mean values of the GMM fit.

Each identified fluorophore position was then subjected to both separate DBSCAN clustering and nearest neighbour analysis steps. The DBSCAN step identified clusters of three fluorophore positions with epsilon = 100 nm and minpts = 3. The nearest neighbour step required that each fluorophore position must have two neighbours, and their centre positions separated by between 6 and 50 nm. In a special case, for Fig. 4b, the nearest neighbour step was adjusted to have a minimum and maximum distance of between 5 and 60 nm, respectively, to capture interblade distances, which were slightly longer than 50 nm. Note that the most probable interblade distance measured at position 103 in a cell is the same as for maximum nearest neighbour distance of 50 nm (Figs. 2e and 4b). Next, clusters of three fluorophore positions passing both steps were segmented. The data were manually z-filtered based on the distribution of raw localizations to isolate only plasma membrane-bound PIEZO molecules. Finally, candidate clusters containing interblade angles of more than 120° were filtered out to eliminate nonspecific trace streaks. For each identified cluster of three molecules, the average interblade distance was directly calculated from fluorophore centre positions.

## PIEZO repeat binding energy calculations
First, the sequence locations of PIEZO repeat domains[5] were isolated from the PIEZO1 (Uniprot entry Q8CD54) and PIEZO2 (Uniprot entry EJ2F22) amino acid sequences. The transmembrane domains were identified using the structures as guidance and split into distinct chains (Supplementary Text). Inter-PIEZO repeat binding energy was calculated using the PDBePISA tool[43]. The total solvation free energy gain upon formation of the interface −ΔG was determined for each binding interface contacting each PIEZO repeat domain with and without the contribution of extracellular loops.

## Structural models
Structural models from cryo-EM were obtained from the Protein Data Bank (PDB), and PDB accession numbers are noted in the article. To generate a trimeric AlphaFold II model, the monomeric E2JF22 prediction was superposed onto the three PIEZO1 subunits of PDB 6B3R. Owing to the lack of confidence in cap placement relative to the PIEZO blades in the AlphaFold model, the CED was removed and not included in structural analyses.

## Control staining and confocal imaging
Cells were prepared exactly as for MINFLUX imaging, except that 1 µM of the cell-permeant Janelia Fluor 549–Halo Ligand (Promega) was added to the tetrazine–Alexa Fluor 647 labelling mix. All images (Extended Data Fig. 2b) were acquired on a Nikon AX confocal microscope with NIS Elements software and the image settings (laser power, gain, resolution, pixel dwell time, ×60 1.4 NA oil immersion plan apochromat objective and pixel dimension settings) were kept the same for all conditions. For all images, the brightness and contrast adjustments were applied uniformly to the entire image.

## Functional verification of tagged PIEZO1 with electrophysiology
For verification of proper function of tagged proteins, TAG-substituted PIEZO1 constructs and pNEU-hMbPylRS-4×U6M15 were transfected at a 1:1 ratio in the presence of 250 µM TCO*K with Lipofectamine 2000 (Thermo Fisher Scientific) into *Piezo1*-knockout HEK293 cells (CRL-3519, American Type Culture Collection) plated onto poly-D-lysine-coated coverslips. For wild-type control experiments, mPIEZO1-IRES-eGFP (plasmid #80925, Addgene) was co-transfected with pNEU-hMbPylRS-4×U6M15 at a 1:1 ratio. Cells were cultured according to the guidelines from the American Type Culture Collection and were verified to be free of mycoplasma using the MycoAlert Mycoplasma Detection Kit (Lonza). Cells were allowed to express for 24–48 h before recording.

For experiments in which the cells were labelled, cells were first washed with DMEM and 20 mM HEPES. All labelling was performed

at room temperature. Next, the cells were blocked in DMEM, 20 mM HEPES and 1 mg ml$^{-1}$ Roche blocking reagent for 5 min at room temperature. After blocking, the cells were labelled with DMEM, 20 mM HEPES, 1 mg ml$^{-1}$ Roche blocking reagent and 4 μM tetrazine–Alexa Fluor 647 for 10 min. The cells were then washed in DMEM, 20 mM HEPES and 1 mg ml$^{-1}$ Roche blocking reagent and exchanged into DMEM and 20 mM HEPES. Labelling was confirmed with visualization on a fluorescence microscope.

Mechanically activated currents from HEK293 *Piezo1*-knockout cells were recorded in whole-cell voltage clamp mode using a Multi-Clamp700A amplifier and DigiData1550 (Molecular Devices) and stored directly and digitized online using pClamp software (version 10.7). Currents were recorded at −80 mV, sampled at 20 kHz and filtered at 2 kHz. Recording electrodes had a resistance of 1.5–3 MΩ when filled with CsCl-based intracellular solution: 133 mM CsCl, 1 mM CaCl$_2$, 1 mM MgCl$_2$, 10 mM HEPES (pH 7.3 with CsOH), 5 mM EGTA, 4 mM Mg-ATP and 0.4 mM Na-GTP. Extracellular bath solution was composed of 133 mM NaCl, 3 mM KCl, 2.5 mM CaCl$_2$, 1 mM MgCl$_2$, 10 mM HEPES (pH 7.3 with NaOH) and 10 mM glucose. Mechanical stimulation was achieved using a fire-polished glass pipette (tip diameter of 3–4 μm) positioned at an angle of 80° relative to the cell being recorded and about 5 μm away from the cell. Displacement of the probe towards the cell was driven by Clampex-controlled piezoelectric crystal microstage (E625 LVPZT Controller/Amplifier, Physik Instrumente). The probe had a velocity of 1 μm ms$^{-1}$ during the ramp phase of the command for forward movement, and the stimulus was applied for a duration of 125 ms. For each cell, a series of mechanical steps in 1-μm increments was applied every 10 s starting with an initial displacement of 5 μm. The step at which the probe tip visibly touched the cell was used as the baseline for determining the apparent threshold (the micrometre above touching the cell at which the first response was observed).

A family of displacement steps (0.5-μm increments) was applied to the cell and mechanically activated currents were recorded usually to 2–3 μm above the threshold response to avoid losing the cell. Yoda1 was diluted from a 20 mM stock solution in DMSO, vortexed aggressively and used within 5 min to avoid precipitation in solution. Yoda1 (10 μM) was applied manually without bath perfusion and cells were exposed for 5 min followed by washout. Families of mechanically activated currents were acquired twice before Yoda1, two to three times in Yoda1 and multiple times during washout.

### Measurement of hypotonic and Yoda1-evoked currents with electrophysiology

For Yoda1 and hypotonic electrophysiology, *Swell1*-knockout HEK293F cells[68] were cultured in FreeStyle 293 medium and maintained as for the Expi293 cells. Cells were verified to be free of mycoplasma using the MycoAlert Mycoplasma Detection Kit (Lonza). Cells were transfected with mPIEZO1-IRES-eGFP (plasmid #80925, Addgene) using the EndoFectin 293 transfection reagent (GeneCopeia) at a concentration of 1 μg ml$^{-1}$. Cells were allowed to express for 48 h and plated onto poly-D-lysine-coated glass coverslips before recording.

Whole-cell currents were recorded using a Axopatch 200B amplifier and Digidata 1440A (Molecular Devices) and analysed with pClamp (version 10.2). Currents were recorded at +80 mV, sampled at 20 kHz and filtered at 2 kHz. Recording electrodes were pulled and polished to an initial resistance of 4–8 MΩ when filled with pipette solution containing the following: 110 mM CsCl, 40 mM CsF, 10 mM EGTA and 20 mM HEPES (pH 7.4).

For Yoda1 experiments, the bath solution contained the following: 140 mM NaCl, 2.4 mM KCl, 10 mM HEPES (pH 7.4), 10 mM D-glucose, 4 mM MgCl$_2$ and 4 mM CaCl$_2$ (307 ± 5 mOsm). Responses were evoked with 50 μM Yoda1 prepared in bath solution.

For osmotic swelling experiments, the bath solution contained 40 mM NaCl, 2.4 mM KCl, 10 mM HEPES (pH 7.4), 10 mM D-glucose, 4 mM MgCl$_2$, 4 mM CaCl$_2$ and 185 mM D-mannitol (310 mOsm ± 5 mOsm).

Responses were evoked with hypotonic bath solution containing 40 mM NaCl, 2.4 mM KCl, 10 mM HEPES (pH 7.4), 10 mM D-glucose, 4 mM MgCl$_2$ and 4 mM CaCl$_2$ (122 ± 5 mOsm). The osmolality of all solutions was measured with a vapour pressure osmometer (Wescor).

### Data visualization and statistical tests

Data were visualized and statistical tests performed in MATLAB (Math-Works) and Prism (GraphPad) software. Molecular structures were visualized in MolStar Viewer (https://molstar.org/viewer/) and Chimera (UCSF) software.

### Reporting summary

Further information on research design is available in the Nature Portfolio Reporting Summary linked to this article.

## Data availability

All data supporting the article, including the raw MINFLUX analysis output for each experimental condition, are provided as source data. Protein structures were obtained from RCSB Protein Data Bank (PIEZO1 6B3R, PIEZO1 7WLU and PIEZO2 6KG7) and the AlphaFold Protein Structure Database (PIEZO1 E2JF22). Raw data and all reagents not commercially available are available from the corresponding author upon reasonable request. Source data are provided with this paper.

## Code availability

The code for iPALM data pre-processing was previously published[26] and is available at https://github.com/gleb-shtengel/PeakSelector. The custom MATLAB code for iPALM data analysis is available at https://doi.org/10.5281/zenodo.8017632. The custom MATLAB code for MINFLUX analysis is available at https://github.com/PatapoutianLab/MINFLUX_Piezo_Analysis.

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

**Acknowledgements** We thank Y. Wang for preliminary conceptualization, C. Wurm, I. Jansen and Abberior Instruments for help with MINFLUX acquisition and instrument and/or software troubleshooting; S. Khuon for technical assistance with iPALM sample preparation; R. MacKinnon, C. Haselwandter and R. Hill for critical reading of the manuscript; S. Hell and J. Keller for sharing cluster analysis code; and members of the Patapoutian laboratory for feedback and discussions. MINFLUX imaging was performed at and supported by the Scripps Research Core Microscopy Facility. iPALM imaging was performed at and supported by the Advanced Imaging Center at the Howard Hughes Medical Institute Janelia Research Campus. The Advanced Imaging Center at Janelia Research Campus is supported by the Howard Hughes Medical Institute and the Gordon and Betty Moore Foundation. This work was supported by NIH grant R01 HL143297. E.M.M. is supported by a George E. Hewitt Foundation for Medical Research postdoctoral fellowship. A.P. is a Howard Hughes Medical Institute Investigator.

**Author contributions** E.M.M. designed and performed all biology and imaging experiments, analysed all data, wrote the MINFLUX analysis code, wrote the manuscript and, together with A.P., conceived the project. R.M.L. wrote the iPALM analysis code, analysed the iPALM data and performed the super-particle fusion. A.G. and A.E.D. performed the electrophysiology experiments. E.M.M. and A.G. performed the binding energy calculations. J.S.A. and M.A.R. helped to prepare the iPALM samples and performed iPALM data acquisition. K.L.M., K.R.S. and J.S.A. pre-processed and analysed the iPALM data. S.C.H. contributed to the MINFLUX imaging experiments and troubleshooting. T.-L.C. and A.P. contributed to project design. A.P. supervised the project. All authors discussed results and contributed to manuscript editing.

**Competing interests** The authors declare no competing interests.

**Additional information**
**Correspondence and requests for materials** should be addressed to Ardem Patapoutian.

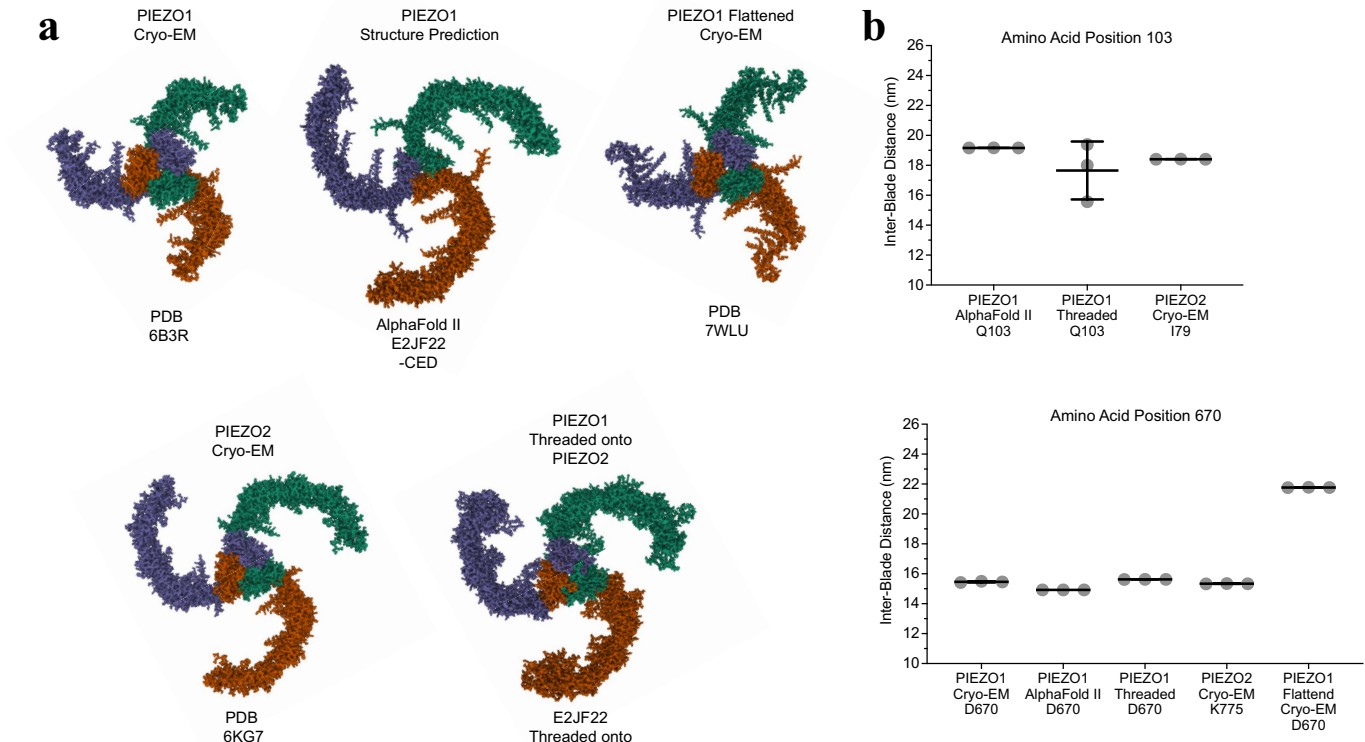

**Extended Data Fig. 1 | Structural models of PIEZOs and inter-blade distance measurements. a**, Scaled molecular renderings of published cryo-EM structures and structural models. Each protomer of the PIEZO trimer is colored blue, orange, and green. **b**, Top, measured inter-blade distances between amino acid position 103 in each protomer of PIEZO1 for models that resolve the last approximately one third of the distal blade. The nearest equivalent amino acid position for PIEZO2 was determined to be isoleucine 79. Bottom, measured inter-blade distances between amino acid position 670 in each protomer of PIEZO1 from all structures shown in **a**. The nearest equivalent amino acid position for PIEZO2 was determined to be lysine 775. Data are presented as mean values ± s.d. For all calculations, *n* = 1 PDB structure.

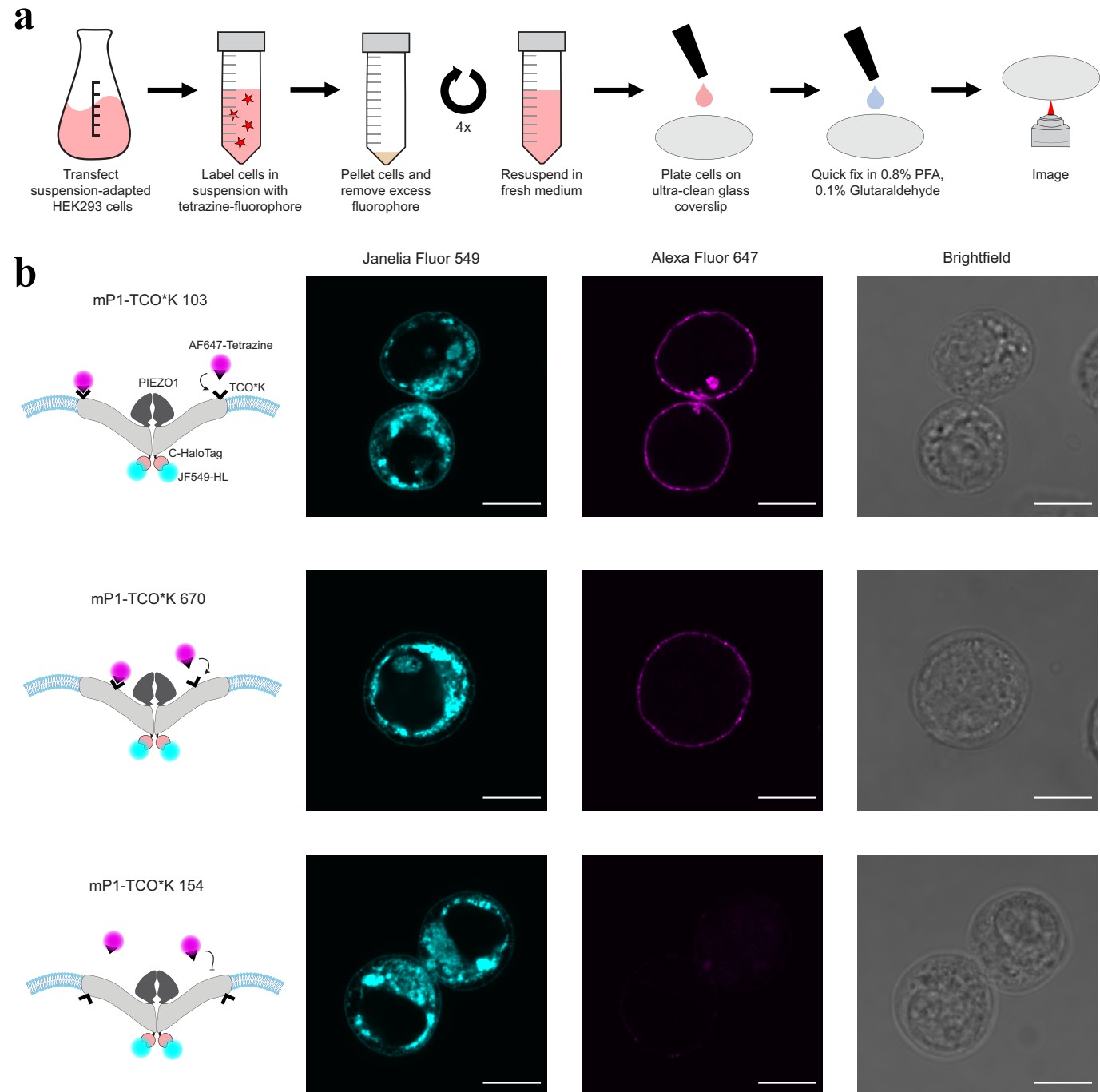

**Extended Data Fig. 2 | Cell preparation and cell-surface labeling of mPIEZO1. a**, Overview of the cell labeling and plating scheme used for both iPALM and MINFLUX imaging. **b**, Cells expressing TCO*K substituted mPIEZO1-C-HaloTag-TwinStrep constructs were labeled live with tetrazine-AF647 as for iPALM and MINFLUX imaging (Methods). The cell-permeant JF549-HaloLigand was used to label the intracellular side of the channel. Tagging and labeling at positions 103 and 670 resulted in specific cell-surface labeling of PIEZO1. Tagging at position 154 on an intracellular loop at a location known to not disrupt function[69] results in cell surface expression of the protein as shown by JF549-HaloLigand labeling, but no labeling with the cell-impermeant tetrazine-AF647. Scale bar = 20 μm. Representative images from $n$ = 20 cells.

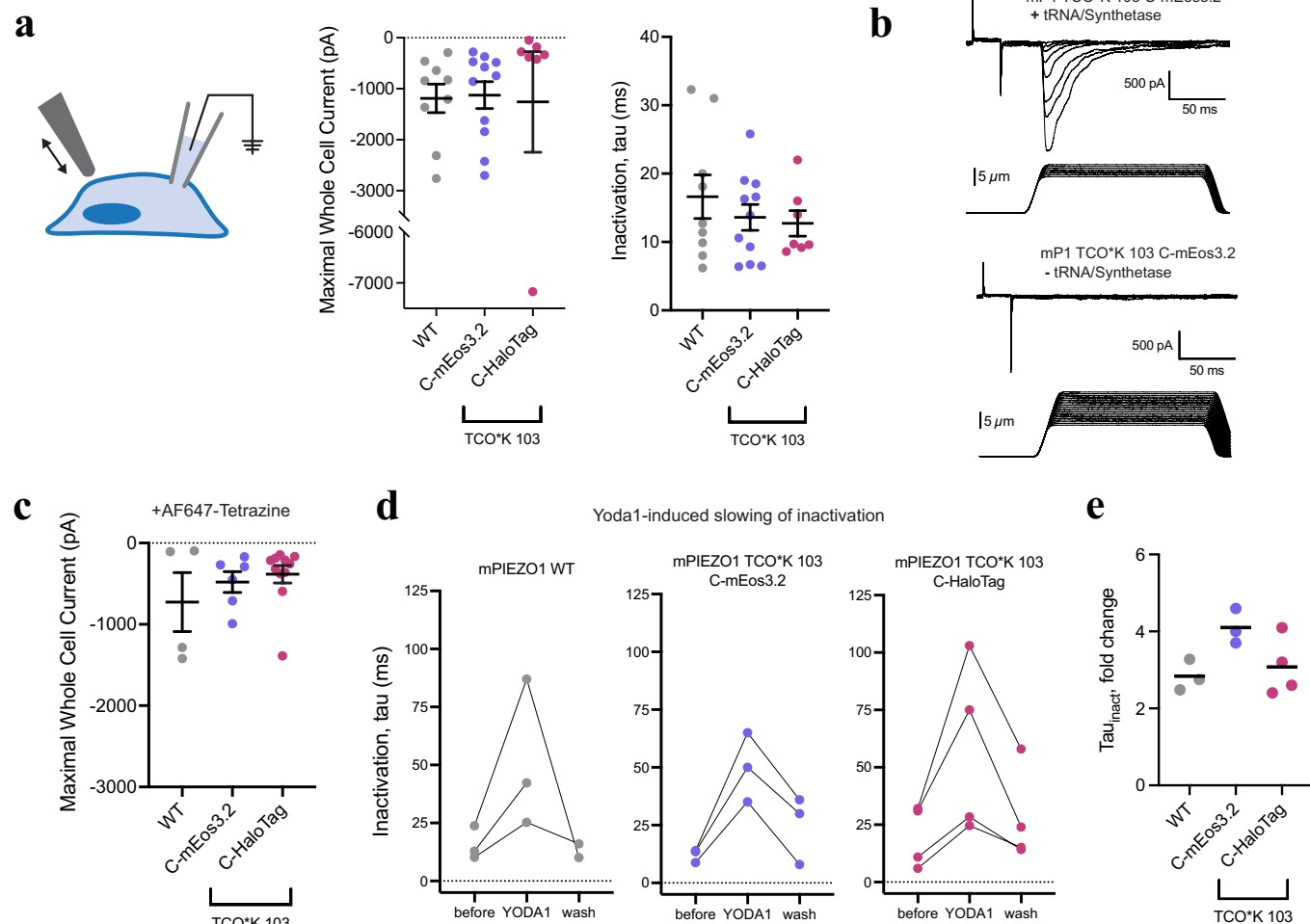

**Extended Data Fig. 3 | Labeled PIEZO1 displays normal electrophysiological properties. a**, Left, schematic of whole cell electrophysiology with cell poking. Middle, maximal whole cell currents evoked by mechanical stimulation. Cells were transfected with mPIEZO1-IRES-eGFP (WT, grey circles, −1189 ± 280 pA, $n = 9$ cells), mPiezo1-TCO*K103-C-mEos3.2-TwinStrepTag (blue circles, −1125 ± 262 pA, $n = 11$ cells), and mPiezo1-TCO*K103-C-HaloTag-TwinStrepTag (red circles, −1259 ± 986 pA, $n = 7$ cells). All cells were co-transfected with the tRNA/Synthetase and cultured in the presence of the TCO*K unnatural amino acid. Right, time constant of inactivation for maximal whole cell currents shown in the middle (WT = 17 ± 3 ms, C-mEos3.2 = 14 ± 2 ms, C-HaloTag = 13 ± 2 ms). All values are mean ± s.e.m. **b**, Representative whole cell currents evoked by mechanical stimulation for TCO*K labeled mPIEZO1 with and without the tRNA/synthetase. **c**, Maximal whole cell currents evoked by mechanical stimulation

from cells labeled with tetrazine-AF647 using the same conditions as in **a**, for cells co-transfected with the tRNA/Synthetase and cultured in the presence of the TCO*K unnatural amino acid. Labeling does not result in a significant change in maximal whole cell current relative to the WT channel (WT = −727 ± 362 pA, $n = 4$ cells; C-mEos3.2 = −480 ± 128 pA, $n = 6$ cells; C-HaloTag = −383 ± 108 pA, $n = 11$ cells). All values are mean ± s.e.m. **d**, Yoda1-induced slowing of channel inactivation. The time constant of inactivation was measured before, during, and after bath application of 10 μM Yoda1. Lines are shown connecting measurements from individual cells. **e**, Quantification of Yoda1-induced slowing of inactivation for each of the three constructs tested. The median fold change in inactivation is shown as a black line (WT = 2.8, $n = 3$ cells; C-mEos3.2 = 4.0, $n = 3$ cells; C-HaloTag = 2.9, $n = 4$ cells).

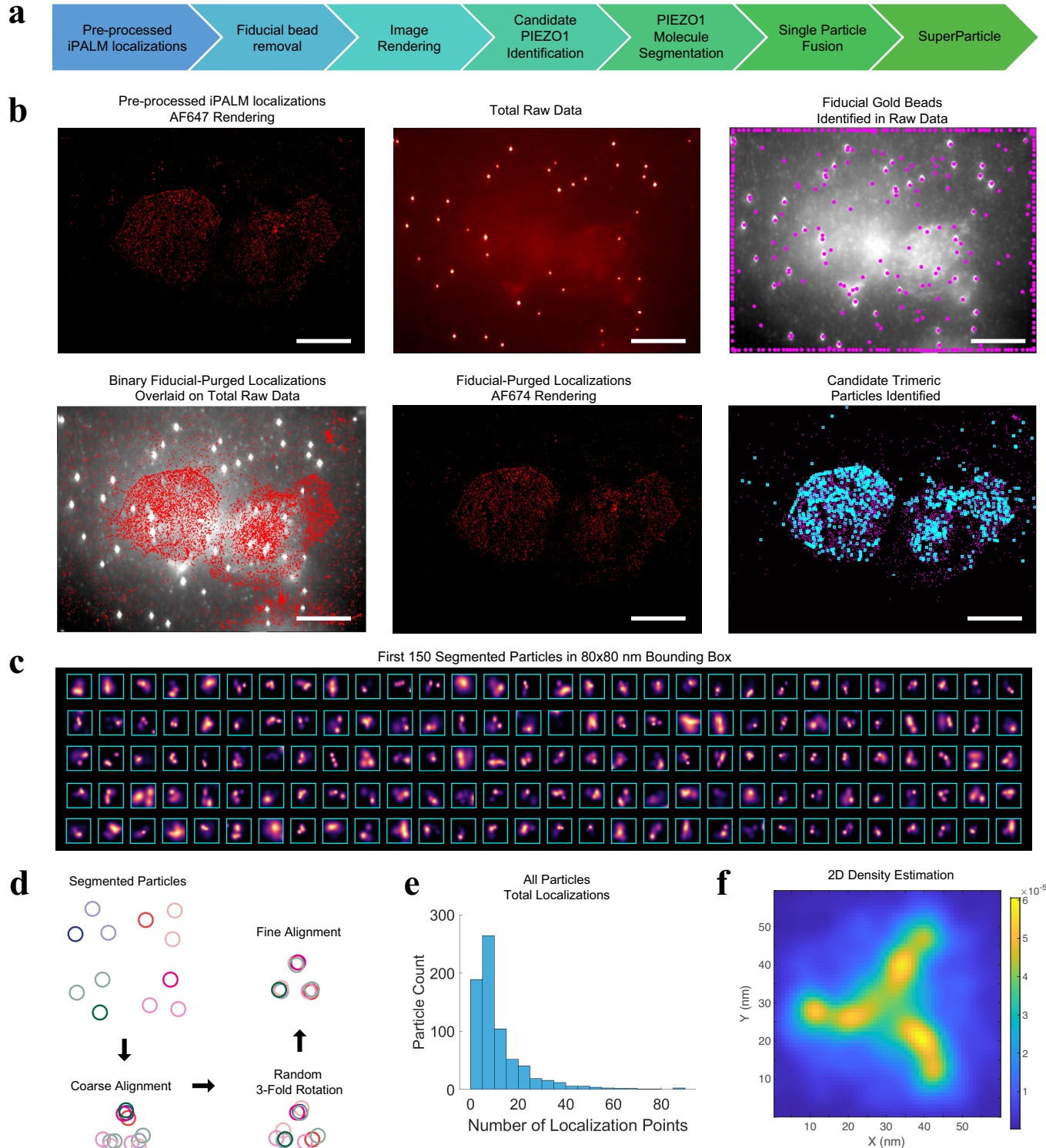

**Extended Data Fig. 4** | See next page for caption.

**Extended Data Fig. 4 | Overview of single-particle segmentation and fusion of iPALM localization data. a**, Workflow of iPALM data processing, particle segmentation, and super-particle fusion. **b**, Representative images (from $n = 5$ cells) showing bead removal and candidate triple labeled PIEZO1 segmentation on pre-processed localizations collected with the 640 nm laser. Top left, pre-processed localizations found at/near the plasma membrane rendered at 4 nm per pixel. Top middle, an image of the total raw photon data collected. The gold fiducial beads are apparent as bright spots since they are constantly emitting photons in each image frame. Top right, fiducial gold beads identified (magenta circles) by the bead removal algorithm from the total raw data image. Bottom left, total raw data overlaid with localizations showing removal of bead-associated localizations. Bottom middle, a 4 nm/pixel rendering of bead fiducial-purged localizations. These localizations are associated with AF647 fluorescence. Bottom right, binary AF647 localizations with candidate PIEZO1 molecules meeting nearest neighbor requirements highlighted with cyan boxes. Scale bars = 20 μm. **c**, The first 150 segmented triple labeled PIEZO1 particles identified from the segmentation algorithm meeting both nearest neighbor and inter-localization separation distance requirements. Each particle is shown in a 40x40 nm cyan bounding box. **d**, Overview of each of the three major steps in the particle fusion algorithm from Heydarian, et al.[28]. **e**, The total number of localizations per identified triple labeled PIEZO1 particle from all iPALM datasets ($n = 5$ imaged cells, $n = 726$ identified particles, $n = 8500$ total localizations). **f**, A 2D density grid estimation of the iPALM super-particle shown in Fig. 1c using a 1x1 nm binned grid and colored by local density.

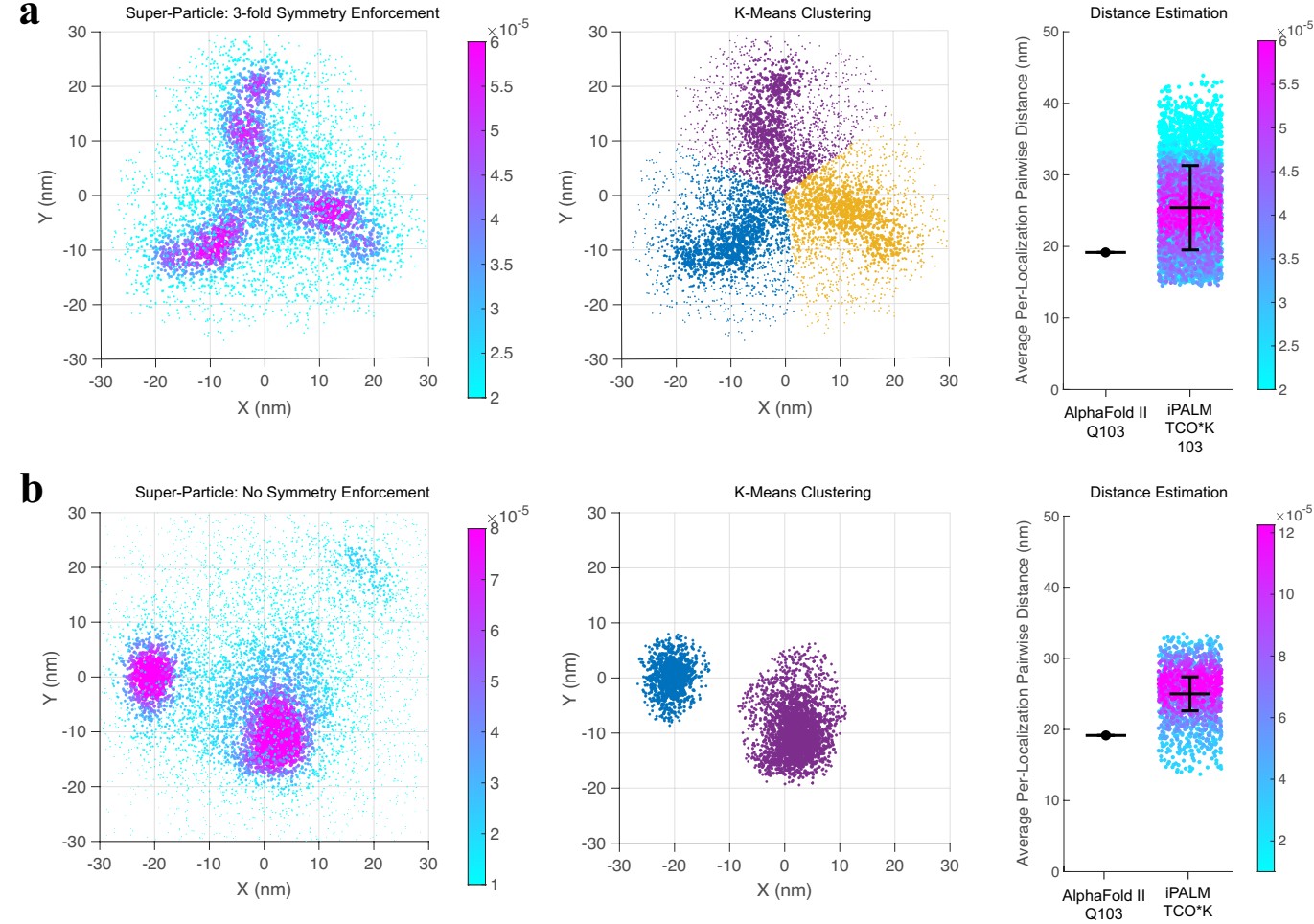

**Extended Data Fig. 5 | Comparison of iPALM Super-Particle Symmetry Enforcement. a**, Left, iPALM super-particle with 3-fold symmetry enforcement, based upon known channel stoichiometry as shown in Fig. 1c ($n$ = 5 imaged cells, $n$ = 726 identified particles, $n$ = 8500 total localizations). Center, $k$-means clustering of blade positions as shown in Fig. 1d. Right, scatter plot of average per-localization inter-blade distances as shown in Fig. 1e. The most probable inter-blade distance was calculated to be 25.4 ± 5.9 nm, compared to 19.2 nm calculated from the AlphaFold II model at position 103. Distances are shown as mean ± s.d. **b**, Left, iPALM super-particle without symmetry using the same dataset as for (**a**). The number of localizations per molecule position is not uniform. Since the registration algorithm tends to match regions of dense localizations, this results in a "hot spot" of localizations[28,29]. Center, $k$-means clustering of the two most dense regions of localizations. The super-particle on the right was thresholded to a local density of $2.75 \times 10^{-5}$ before clustering. Right, scatter plot of average per-localization inter-blade distances. The most probable inter-blade distance was calculated to be 25.0 ± 2.4 nm, compared to 19.2 nm calculated from the AlphaFold II model at position 103. Distances are shown as mean ± s.d.

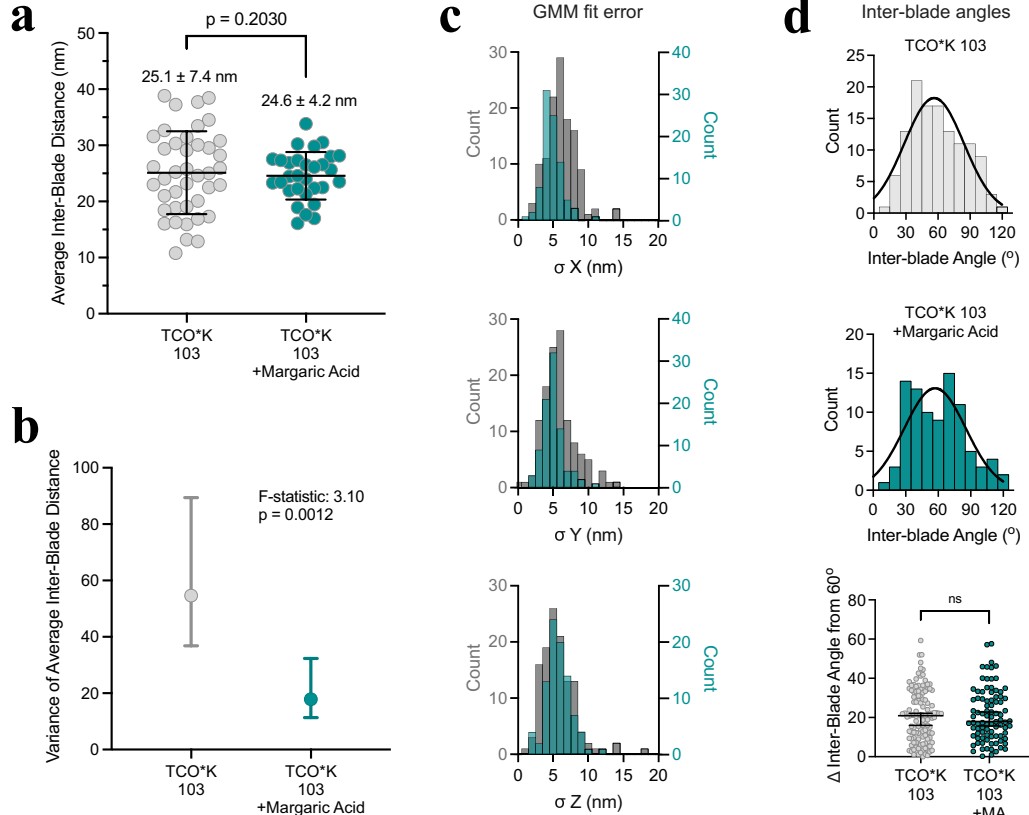

**Extended Data Fig. 6 | Saturated Fatty Acid Enrichment of the Plasma Membrane. a**, Scatter plot of inter-blade distances at amino acid position 103 with (green circles, $n = 30$ molecules, $n = 3$ cells) and without (grey circles, $n = 41$ molecules, $n = 5$ cells) enrichment of the plasma membrane with 300 μM margaric acid. Kolmogorov-Smirnov: $P = 0.2030$ and $D = 0.2569$. Distances are shown as mean ± s.d. **b**, Variance and 95% confidence interval of average inter-blade distances at position 103 with and without margaric acid enrichment from the scatter plot data in (**a**). F-test of equality of variances: $P = 0.0012$, F-statistic = 3.10). **c**, Histogram of Gaussian mixture model fit error for all identified triple labeled PIEZO1 molecules at position 103 with (green) and without (grey) margaric acid enrichment. Bin width = 1 nm. **d**, Top, histograms of inter-blade angles for identified triple labeled PIEZO1 channels at position 103 without enrichment ($56.4 ± 27.8$ nm, $R^2 = 0.84$) and with margaric acid enrichment ($57.1 ± 28.4$ nm, $R^2 = 0.71$). Inter-blade angles were binned by 10 nm and fit with a Gaussian. Values are mean ± s.d. Bottom, scatter plot of the change in inter-blade angles from symmetric (60°). Mann-Whitney: $P = 0.9097$, $U = 5484$. Error bars are shown as median and 95% confidence interval. All statistical tests are two-tailed.

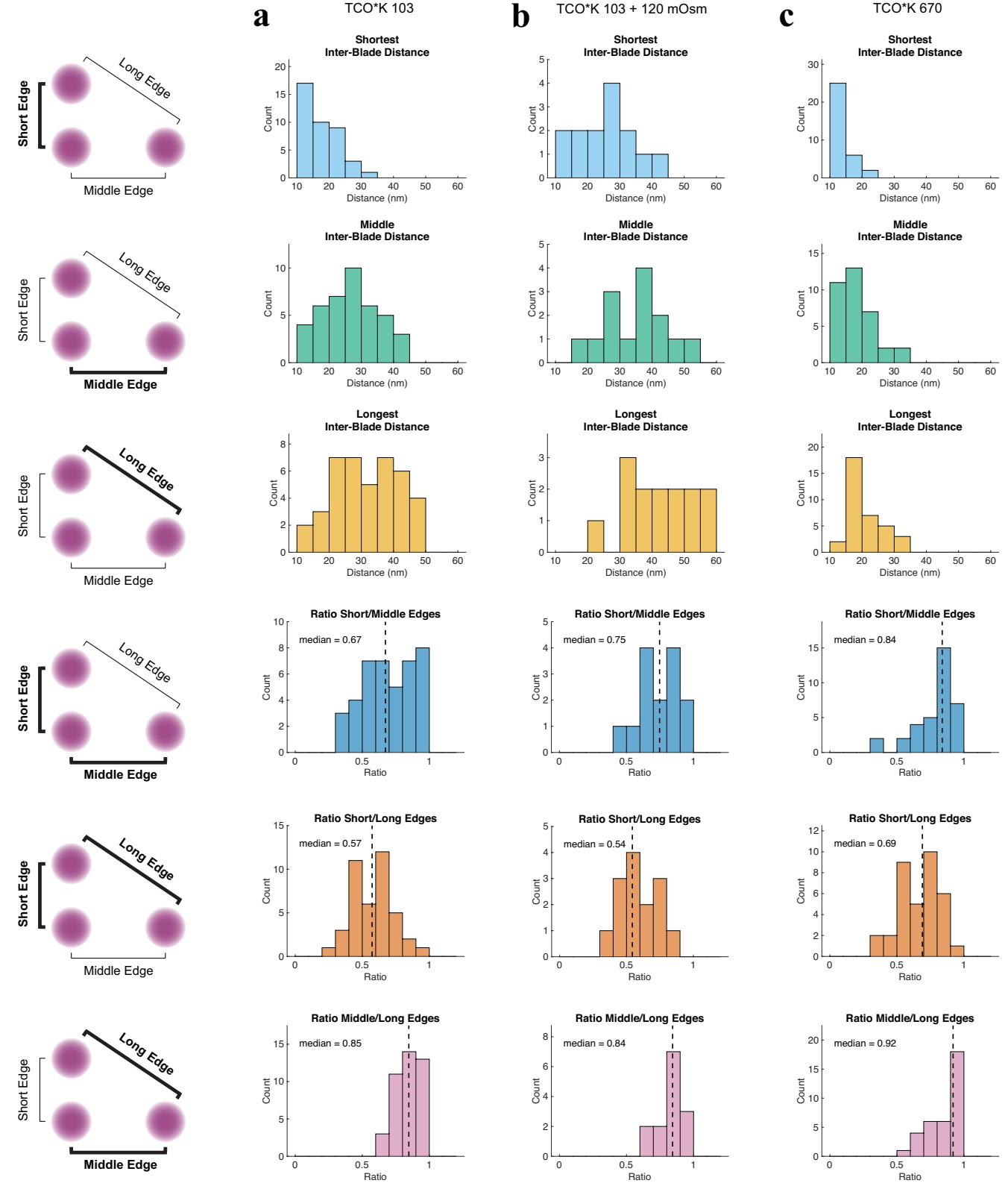

**Extended Data Fig. 7 | Analysis of PIEZO1 Blade Cooperativity.** Histograms of inter-blade distances for each edge of the triangle formed by fluorophore positions of isolated triple labeled PIEZO1 molecules and the ratios of each of these edges. **a**, Inter-blade distances and edge ratios for TCO*K 103 in the absence of stimulus (from Fig. 2e). **b**, Inter-blade distances and edge ratios for TCO*K 103 with a 120 mOsm hypotonic shock stimulus (from Fig. 4b). **c**, Inter-blade distances and edge ratios for TCO*K 670 in the absence of stimulus (from Fig. 3c).

# Reporting Summary

## Statistics

For all statistical analyses, confirm that the following items are present in the figure legend, table legend, main text, or Methods section.

| n/a | Confirmed | |
|---|---|---|
| ☐ | ☒ | The exact sample size (*n*) for each experimental group/condition, given as a discrete number and unit of measurement |
| ☐ | ☒ | A statement on whether measurements were taken from distinct samples or whether the same sample was measured repeatedly |
| ☐ | ☒ | The statistical test(s) used AND whether they are one- or two-sided *Only common tests should be described solely by name; describe more complex techniques in the Methods section.* |
| ☒ | ☐ | A description of all covariates tested |
| ☒ | ☐ | A description of any assumptions or corrections, such as tests of normality and adjustment for multiple comparisons |
| ☐ | ☒ | A full description of the statistical parameters including central tendency (e.g. means) or other basic estimates (e.g. regression coefficient) AND variation (e.g. standard deviation) or associated estimates of uncertainty (e.g. confidence intervals) |
| ☐ | ☒ | For null hypothesis testing, the test statistic (e.g. *F*, *t*, *r*) with confidence intervals, effect sizes, degrees of freedom and *P* value noted *Give P values as exact values whenever suitable.* |
| ☒ | ☐ | For Bayesian analysis, information on the choice of priors and Markov chain Monte Carlo settings |
| ☒ | ☐ | For hierarchical and complex designs, identification of the appropriate level for tests and full reporting of outcomes |
| ☒ | ☐ | Estimates of effect sizes (e.g. Cohen's *d*, Pearson's *r*), indicating how they were calculated |

*Our web collection on statistics for biologists contains articles on many of the points above.*

## Software and code

Policy information about availability of computer code

| | |
|---|---|
| Data collection | iPALM data was collected with custom LabView software (LabView 2010, National Instruments), as described in Shtengel, G. et al. 2009. MINFLUX data was collected with Imspector software with MINFLUX drivers (version 16.3, Abberior Instruments). Confocal images were collected with NIS-Elements software (version 5.40.01, Nikon). Electrophysiology data was collected with pClamp software (version 10.2 and 10.7, Molecular Devices). PIEZO repeat binding energy calculations were performed using PDBePISA software (version 1.52, https://www.ebi.ac.uk/pdbe/pisa/). |
| Data analysis | iPALM data pre-processing was performed using PeakSelector software (https://github.com/gleb-shtengel/PeakSelector). iPALM data analysis, particle fusion, and MINFLUX data analysis was performed using custom software written in MATLAB (version R2021b, MathWorks). Confocal and brightfield images were analyzed using Fiji (version 2.30, https://fiji.sc/). DNA sequences were created and analyzed in SnapGene (Version 6.2, Dotmatics). Data visualization and statistical tests were performed with MATLAB 2021 (version R2021b, MathWorks) and Prism (version 9.5, GraphPad). Visualization of localizations in Fig. 2 were performed using ParaView (version 5.10, kitware). Molecular structures were visualized using MolStar viewer (https://molstar.org/viewer/) and Chimera software (version 1.15, UCSF). Graphics were created using Adobe Illustrator (version 2023, Adobe). Code for iPALM data pre-processing wass previously published and available at https://github.com/gleb-shtengel/PeakSelector. Custom MATLAB code for iPALM data analysis is available at https://doi.org/10.5281/zenodo.8017632 and custom MATLAB code for MINFLUX analysis is available at https://github.com/PatapoutianLab/MINFLUX_Piezo_Analysis. |

For manuscripts utilizing custom algorithms or software that are central to the research but not yet described in published literature, software must be made available to editors and reviewers. We strongly encourage code deposition in a community repository (e.g. GitHub). See the Nature Portfolio guidelines for submitting code & software for further information.

## Data

Policy information about availability of data

All manuscripts must include a data availability statement. This statement should provide the following information, where applicable:

- Accession codes, unique identifiers, or web links for publicly available datasets
- A description of any restrictions on data availability
- For clinical datasets or third party data, please ensure that the statement adheres to our policy

All data including the raw MINFLUX analysis output for each experimental condition are available online as separate Excel files for each figure. Protein structure data was obtained from RCSB Protein Data Bank (PIEZO1 6B3R: https://doi.org/10.2210/pdb6B3R/pdb, PIEZO1 7WLU: https://doi.org/10.2210/pdb7WLU/pdb, and PIEZO2 6KG7: https://doi.org/10.2210/pdb6KG7/pdb) and the AlphaFold Protein Structure Database (PIEZO1: https://alphafold.ebi.ac.uk/entry/E2JF22). Unprocessed data are available from A.P. upon request.

## Human research participants

Policy information about studies involving human research participants and Sex and Gender in Research.

| | |
|---|---|
| Reporting on sex and gender | N/A |
| Population characteristics | N/A |
| Recruitment | N/A |
| Ethics oversight | N/A |

Note that full information on the approval of the study protocol must also be provided in the manuscript.

# Field-specific reporting

Please select the one below that is the best fit for your research. If you are not sure, read the appropriate sections before making your selection.

☒ Life sciences   ☐ Behavioural & social sciences   ☐ Ecological, evolutionary & environmental sciences

For a reference copy of the document with all sections, see nature.com/documents/nr-reporting-summary-flat.pdf

# Life sciences study design

All studies must disclose on these points even when the disclosure is negative.

| | |
|---|---|
| Sample size | At least three samples were measured for each condition to account for variability in samples and sample preparation. For MINFLUX imaging, we imaged a minimum of three cells, but in some cases, we continued to acquire data until enough molecules were identified to capture the distribution reflecting the apparent mechanical behavior of the imaged region of the molecule. For iPALM imaging, we determined a sample size of 5 cells based upon the estimated number of molecules needed for a fused superparticle of sufficient resolution, given the number of apparent triple-labeled molecules from the first dataset. |
| Data exclusions | For imaging of PIEZO1 cell membranes, data outside the plasma membrane was excluded based on either calculated distance to fiducial markers on the coverslip or the distribution of localizations in the z-plane. For MINFLUX imaging of PIEZO1, those identified trimeric molecules which had an inter-blade angle >120° were excluded as these were primarily due to linear "streaks" in the data, likely from freely diffusing molecules which are universally present in various sample types (including samples of different molecules and labels) in our hands. |
| Replication | At least three biological and experimental replicates were performed for each experiment. All attempts at replication were successful. |
| Randomization | This study did not allocate experimental units to groups, and so no randomization was required for any experiment reported. |
| Blinding | Blinding was not relevant to this study because all data was analyzed using the same methods. |

# Reporting for specific materials, systems and methods

We require information from authors about some types of materials, experimental systems and methods used in many studies. Here, indicate whether each material, system or method listed is relevant to your study. If you are not sure if a list item applies to your research, read the appropriate section before selecting a response.

## Materials & experimental systems

| n/a | Involved in the study |
|---|---|
| ☒ | ☐ Antibodies |
| ☐ | ☒ Eukaryotic cell lines |
| ☒ | ☐ Palaeontology and archaeology |
| ☒ | ☐ Animals and other organisms |
| ☒ | ☐ Clinical data |
| ☒ | ☐ Dual use research of concern |

## Methods

| n/a | Involved in the study |
|---|---|
| ☒ | ☐ ChIP-seq |
| ☒ | ☐ Flow cytometry |
| ☒ | ☐ MRI-based neuroimaging |

## Eukaryotic cell lines

Policy information about cell lines and Sex and Gender in Research

| | |
|---|---|
| Cell line source(s) | Expi293 cells were obtained from ThermoFisher Scientific. Swell1-knockout cells were of the Freestyle HEK293-F cell line, originally obtained from ThermoFisher Scientific, and modified as described in Kefauver, et al. 2018. Piezo1 KO HEK293 cells were generated as described in Lukacs, et al. 2015 and are deposited with ATCC (ATCC CRL-3519). |
| Authentication | Commercially available cell lines were authenticated by the supplier. Knockout of the genes encoding Swell1 (LRRC8A, LRRC8B, LRRC8D, and LRRC8E) in the Swell1-KO cells were verified in Kefauver, et al. 2018. Successful knock-out of Swell1 genes was determined by PCR genotyping and Sanger sequencing targeted regions for frameshift mutations and verified by mass spectrometry analysis. Knock-out of PIEZO1 was verified using PCR genotyping and Sanger sequencing of PIEZO1 alleles. |
| Mycoplasma contamination | All cell lines tested negative for mycoplasma contamination using the MycoAlert® Mycoplasma Detection Kit (Lonza). |
| Commonly misidentified lines (See ICLAC register) | None |

