## [Peer Review File · Nature]

Manuscript Title: Direct Observation of the Conformational States of PIEZO1

Reviewer Comments & Author Rebuttals

Reviewer Reports on the Initial Version:

Referees' comments:

Referee #1 (Remarks to the Author):

In this manuscript Mulhall and colleagues use super-resolution imaging to directly observe conformational dynamics of the blade regions of individual Piezo1 ion channels in a cell, both at rest and in response to osmotic stimulation and gating modifiers. They make the surprising finding that Piezo blades are substantially expanded at rest (as compared to structures obtained with cryo-em) and that the degree of expansion varies dramatically along the length of the blades. They further show that blade expansion is correlated with channel activation, which suggests it may underly the mechanism of mechanosensing. This study is technically highly innovative and provides deep insight into the molecular mechanisms of Piezo function. I am extremely enthusiastic about this work and have only some suggestions to improve what is already an outstanding manuscript:

Specific comments:

1. Lines 316-319: The authors should discuss more precisely how exactly the flexibility of blades (floppy ends) impacts Piezo channel function. Specifically, Haselwandter and MacKinnon recently formulated how changes in dome curvature (R), cross-sectional area (A), and bending modulus (K) impact the tension-activity relationship of Piezos (Figure 4 of <https://www.pnas.org/doi/10.1073/pnas.2208034119>). How do the authors believe each of these parameters, and consequentially tension-response curves, are affected/distorted?
2. Do the collected data allow to analyze the asymmetry/cooperativity of blade extensions? For this, the authors could analyze inter-blade (really: inter-fluorophore) distances for each individual channel, then calculate individual inter-blade distance ratios, and finally histograms of ratios for all channels. Inter-blade distance ratios of ~ 1 would indicate a high degree of cooperativity, whereas values deviating from 1 inform to what extent blades can move independently.
3. The authors suggest that the varying blade flexibility is due to differential binding energies between individual blade domains. The authors may be able to test this hypothesis experimentally by altering the binding strength with point mutations (weakening) or double-cysteine crosslinking (strengthening). The latter may even allow for acute and reversible manipulation, which would be an elegant extension of their work.
4. Line 131: An alternative mechanism for the observed heterogeneity in blade expansion may be local (nm scale) differences in membrane properties, such as topography and bending modulus.

After all, these measurements are performed in cells. The authors should consider adding this interpretation to their manuscript.

5. Line 328: The claim that conformation dynamics of individual membrane proteins have not been previously observed neglects all prior work using single-molecule FRET (for example <https://doi.org/10.1038/s41589-019-0240-7>). The authors better tone down their statement accordingly.

6. Line 101: I find the expression “roughly in plane with the plasma membrane” confusing, because the membrane is predicted to be curved into the footprint. Please try improving the clarity.

7. Figures 1A and S1A: the alpha-fold Piezo subunits are colored as if they were not domain swapped. Please correct.

8. Figures S3a, b, and e: The y-axes should be rescaled to improve data visualization.

9. Figures 3e, f, and 4 a, b, c, d: the figure legends do not match the order of figure panels.

Referee #2 (Remarks to the Author):

The manuscript by Mulhall et al. describes an in-depth analysis of mechanical gating properties in cellular environment of the Piezo1 channel containing genetically encoded fluorescence. By using nanoscopic fluorescence imaging and applying stimuli to cells the authors could directly observe the conformational changes of the Piezo1 channel blades and measure how blade expansion underlies channel activation. They found that the cellular environment can shape the structure of the Piezo1 expanding blades and show how blade expansion affects the channel gating and structure indicating the importance of the local membrane environment for determination of the structural properties of the channel. Compared to structural models of the isolated Piezo1 protein, single-molecule imaging of the channel in cellular environment enabled the authors to detect a dynamic of local protein conformations, which led them to conclude that subunits in membrane proteins could be conformationally heterogeneous. This is an interesting conclusion suggesting that improved labeling methods could further advance the research aiming to resolve the structure of membrane proteins in their native cellular environment. Overall, this study presents an important exploration of the structural dynamics of the Piezo1 channel protein. The manuscript is well written, and the authors present their results concisely. Nevertheless, the authors should consider and address the following comments that can help to significantly strengthen the interpretation of the results in this otherwise very worthwhile study.

Major:

1. Line 130-131: “ ... the standard deviation of inter-blade distances is greater than the experimental localization error, supporting the hypothesis that the spread of distances is driven by intrinsic blade flexibility.” Given the standard deviations of ± 4.0 nm and ± 6.0 nm for the inter-blade distance change from detergent to membrane-bound form (line 916) and cell membrane exposed to GsMTx-4

to AlphaFold II structure (line 920), respectively, compared to the median fluorophore localization error in X, Y, and Z of 5.9, 5.8, and 5.3 nm, this hypothesis does not seem to be well supported by the data shown. To strengthen this hypothesis the authors should perform additional MINFLUX imaging experiments of Piezo1 in cells treated with polyunsaturated fatty acids and margaric acid, which decrease and increase membrane stiffness, respectively (Romero et al. (2019) Nat. Commun. 10: 1200) and determine the inter-blade distances under these conditions that will give them a greater opportunity for observing differences in intrinsic blade flexibility. These experiments would also help to support their hypothesis that the plasma membrane exerts sufficient force to bend the blades of PIEZO1 (line 107-109).

2. Line 156-165: The authors suggest that GsMTx-4 inhibits channel activity by compacting the blades of Piezo1 resulting from reduction of membrane bending stress due to GsMTx-4 interaction with the lipid bilayer acting as a “mobile reserve” of membrane material. However, GsMTx-4 penetration into the membrane bilayer shifts between shallow and deep penetration depending on bilayer tension (Gnanasambandam et al. Biophys. J. 112(1): 31-45, 2017), which distorts the distribution of membrane tension and thus, to a different extent, affects the transfer of force from the bilayer to an inherently mechanosensitive ion channel like Piezo1. Since the standard deviation for the inter-blade distance change from cell membrane exposed to GsMTx-4 to AlphaFold II structure was ± 6.0 nm and thus it is within the range of the median fluorophore localization error (see the previous comment), the concentration of 20 μ M GsMTx-4 applied in MINFLUX cell experiments might not have been adequate to detect the effect suggested by the cartoon shown in Fig. 2g. MINFLUX imaging of Piezo1 reconstituted into a well-defined lipid environment of lipid vesicles (Lin et al. (2019) Nature 573: 230-234) would allow for better distinction of the GsMTx-4 effect on Piezo1 blade extension by applying it at low and high concentrations. Although desirable, I leave it to the authors to decide if they wish to include these experiments in the revised manuscript. They should, nevertheless, comment on the appropriateness of choosing the concentration of 20 μ M GsMTx-4 in their study.

3. Line 178: “PIEZO1 is spontaneously active in the absence of lateral membrane tension in a cell ...”. Spontaneous activity of Piezo1 has been observed in membrane patches in patch clamp experiments and therefore, this is not necessarily true for a cell. This is because the resting tension in the dome of a membrane patch ranges from 1–4 pN/nm (mN/m) depending on the recording mode (Suchyna et al. (2009) Biophys. J. 97(3): 738-47), which is sufficient to induce spontaneous activity of Piezo1.

Minor:

1. Line 136: “ ... blade expansion in a cell is due ‘to’ membrane bending forces.”

2. Line 175-176: “ ... exerted primarily through the plasma membrane rather than through tethering to the cytoskeleton or extracellular matrix.” The papers by Cox et al. (Nat. Commun. 7: 10366, 2016) and Syeda et al. (Cell Rep. 17: 1739–1746, 2016) reporting first the activation of Piezo1 by forces from the plasma membrane should be cited here.

3. Line 240: “... in response ‘to’ hypo-osmotic shock.”

4. Line 328-329: “The data described here show, to our knowledge, the first time that conformational dynamics of individual membrane proteins have been directly observed with fluorescence microscopy at the level of single molecules.” This statement is incorrect. MINFLUX fluorescence imaging of Nup96, a nuclear pore complex protein was done previously as reported in reference 31 (Schmidt et al. (2021) Nat Commun 12: 1478).

5. Fig. 1d, line 852-853: Blue and yellow clusters on the bottom right, do not match the blue and

yellow clusters on the top after 90o rotation.

Referee #3 (Remarks to the Author):

Do I feel the results presented are of immediate interest to many people in my own discipline or to people from several disciplines?

Yes, for a very interesting structural investigation applying these types of microscopy and analysis on a complex outside of the standard samples used for technique development. I think this is the first example in an original biological experiment of observing conformational states and deducing other properties using this type of microscopy, which can access states more native than is possible in single-particle cryo-EM.

More clarity is needed about the analysis, and therefore the results and their interpretation. If these can be improved, publication of this manuscript will advance the use of these techniques and improve the understanding of their potential.

Statistics: The authors have checked n/a for null-hypothesis testing, but have use K-S tests with significance values. The checklist indicates that the test statistic (often D) should be included in the presentation of results, but is missing.

Some uncertainties on measurements and comparisons have been included, but more would be helpful.

Abstract

I am not clear on how surprising it is that expansion should vary along the blade, especially with greater flexibility further from the centre.

These methods will serve as a framework for structural analysis of membrane proteins...

The application of these methods in an experiment like this is indeed an advance, but the methods themselves are each not a novel development, although bespoke elements have been used in the analysis.

Segmentation and particle fusion

...promoting a 3-fold symmetry by rotating each initially aligned particle by a random integer factor of $2\pi/3$...

It should be noted that the very clear 3-fold symmetry observed in the super-particle of Fig. 1c and used in other iPALM results is ensured by this "symmetry promotion", if particles do share a preferred orientation to align on (but they do not need to be 3-fold symmetric). It is illustrated in Ext. Data Fig. 4d, but this "symmetry-promoted" term should be used in the text and Fig. 1c (and possibly elsewhere) to avoid the reader thinking that a super-particle without this symmetry promotion would have this appearance.

It would be good to consider the effect of this symmetry promotion on the elongated density patterns that are noted in the super-particle.

A visualisation of the super-particle before symmetry promotion (perhaps similar to Fig. 1c) would be informative.

Given this bias on the super-particle, I am not clear on the significance of the inter-blade distances measured within it. These are distances between localisation positions in rotated particles in $\sim 2/3$ cases, not measurements between localisations in aligned particles without symmetry promotion. In particular, the effect on uncertainties of these measurements within the symmetry-promoted particle needs considering. Further explanation of the use of these measurements in this light would be helpful.

Using intra-complex pair-wise distances between localisations or clusters without alignment and averaging may be an alternative.

Single-molecule imaging of Piezo1 in a cell membrane

...at least several nanometres more extended...

Perhaps 'at least' is not needed here. 'Most probable' appears right (given the caveats above) from Fig. 1e, but an uncertainty value would be helpful (also considering the caveats above). The authors could consider the use of the functions in Churchman (Biophys. J. 90, 668-671 (2006)) for estimating mean and variability.

MINFLUX data acquisition

Laser powers are given as percentages, but this then needs more information about the lasers (e.g. what would maximum power have been?)

'Mean standard deviation for stabilization' needs explaining. And is then corrected for in some way?

MINFLUX data analysis

...the first DBSCAN...required 5 neighbours for the core point...

This should be ...for 'a' core point...

'candidate clusters containing blade angles > 120 degrees were filtered out to eliminate non-specific trace streaks'

What are these streaks? Is natural variability of the complexes actually being removed?

It is better in the case of the MINFLUX data that the centers of the clusters are used for direct intra-complex distance measurements.

What are the centre positions for a cluster? Centroid? Centre of mass?

Blade flexibility is driven by a differential energetic stability

'Notably, the s.d. of inter-blade distances was 1.6 times less at position 670...'

For this to appear notable, more explanation might be needed. Position 670 is closer to the fixed centre than position 103, and a smaller variation of its position relative to the centre might be expected, even if each repeat has similar flexibility. - Or those compounded flexibilities along the blade might be expected to result in a wider range of positions at position 103.

There are no uncertainties on the bending distances: 'the blade at proximal repeat F is bent by only 2.8 nm, compared to 6.1 nm at distal repeat.'

This affects the significance of this difference, for which an uncertainty value would also help.

The panel letters in Fig. 3 do not match the caption.

The illustrations in Fig. 3f are not very informative. A density plot showing the structure smeared out by the range of angles might be more informative.

"The blades were on average more symmetrical" at repeat F vs. repeat I.

The variabilities are large, and a statistical assessment of the significance of this difference would be helpful.

Activation of Piezo1 by blade extension

Projected area is used only for the test of osmotic shock response of Piezo1, not inter-blade distance, as is used everywhere else. Why is this?

The panel letters in Fig. 4 do not match the caption.

'Distances are shown' should be 'Areas are shown' at one point in Fig. 4.

...position 103 ... incubated with ... Yoda1, the average inter-blade distances increased by 2.08 nm (Fig. 4f).

This ideally needs an uncertainty value and the statistical information ($p=0.393$) should be brought closer to it.

The corresponding result for position 670, 1.96 nm, would also ideally have an uncertainty on it.

In the light of the weak $p=0.393$ result for position 103 and the absence of uncertainties on the measurements, the assertion that Yoda1 uniformly expands the blade seems a bit strong, currently.

Also perhaps the highlight of the power of the technique to accurately observe nm-scale molecular movements - this does seem impressive at times in the manuscript, but the inconclusive measurements of position 670 in this case do not show that.

Discussion

'The extent of blade expansion is driven by a gradient of blade flexibility' - Such a gradient does not seem clear from the imaging, as opposed to possibly constant flexibility along the blade resulting in greater movement at distal regions. It may arise from the binding energy calculation, though.

'We also show that the extent of Piezo1 activation... appears to be explained by the degree of blade expansion.'

Is the data more tentative than this? The authors did put forward a theory on why the degree of channel activation appeared similar for chemical and mechanical modulators, but not one that was covered by the experiments.

'Correlation between the extent of blade extension and channel activity'

The similarity of activations has been presented results, but not so much this correlation at different blade extension.

'We show how the differential flexibility of the blades is determined by the binding strength of Piezo repeat domains.'

This currently seems a circular argument, since the imaging does not seem conclusive about a flexibility gradient (as opposed to constant but compounded flexibility along the blade), and the binding strengths are currently the clearest values suggested to produce such a gradient.

'These properties are likely dictated by graded blade flexibility.'

Again, why would the flexibility need to be graded for this property?

'We expect that these methods will provide a foundation for the use of fluorescence nanoscopy for single-molecule structural biology.'

Yes, iPALM, MINFLUX and Heydarian's particle alignment (albeit with the symmetry promotion) are the foundation. And this is an exciting application where these techniques have been brought to bear on a genuine biological question. If they can be clearer about the results (given the analysis methods) and their interpretation, then the authors have shown the potential of these techniques in original structural biological experiments for the first time, to my knowledge.

Author Rebuttals to Initial Comments:

We would like to thank the reviewers for these great suggestions and for the constructive feedback. Please find our specific responses below.

Referees' comments:

Referee #1 (Remarks to the Author):

In this manuscript Mulhall and colleagues use super-resolution imaging to directly observe conformational dynamics of the blade regions of individual Piezo1 ion channels in a cell, both at rest and in response to osmotic stimulation and gating modifiers. They make the surprising finding that Piezo blades are substantially expanded at rest (as compared to structures obtained with cryo-em) and that the degree of expansion varies dramatically along the length of the blades. They further show that blade expansion is correlated with channel activation, which suggests it may underly the mechanism of mechanosensing. This study is technically highly innovative and provides deep insight into the molecular mechanisms of Piezo function. I am extremely enthusiastic about this work and have only some suggestions to improve what is already an outstanding manuscript:

Specific comments:

1. Lines 316-319: The authors should discuss more precisely how exactly the flexibility of blades (floppy ends) impacts Piezo channel function. Specifically, Haselwandter and MacKinnon recently formulated how changes in dome curvature (R), cross-sectional area (A), and bending modulus (K) impact the tension-activity relationship of Piezos (Figure 4 of <https://www.pnas.org/doi/10.1073/pnas.2208034119>). How do the authors believe each of these parameters, and consequentially tension-response curves, are affected/distorted?

In Haselwandter and MacKinnon's models, the extent of expansion is determined by several mechanical properties including the PIEZO1 protein intrinsic radius of curvature R_0 , intrinsic projected area A_P , bending rigidity K_P , and lipid bilayer bending modulus K_b . Our measurements substantiate previously reported values of R_0 and A_P from structures (Fig. 2f), but we show that the extent of membrane-mediated blade expansion is large and that the flexibility of the blades is high, indicating that the overall protein bending rigidity K_P is exceptionally low. An important consequence of low K_P is that less external force is required to increase channel activity, which confers a high sensitivity to membrane tension¹⁻³. Consistent with our data, these models predict that at least part of PIEZO1 might be similarly flexible as a lipid bilayer, which means that thermal fluctuations alone can result in substantial deformations of the shape of PIEZO1. Specifically, their "mean curvature model" suggests that thermal energy on the order of $k_B T$ can readily deform PIEZO, with the fluctuations overall skewing towards flatter, more expanded shapes. This type of skewing appears to be consistent with the data shown in Fig. 1e and Fig. 2f. The effect of the lipid bilayer bending modulus K_b on PIEZO1 is also predicted by these models and suggest that increasing K_b should both decrease the threshold for gating and increase the degree of expansion by increasing the membrane bending force. Increasing K_b should also consequently decrease the magnitude of blade displacement by random thermal motion⁴. In a response to reviewer #2, we also tested the effect of modulating K_b by increasing membrane stiffness with the saturated fatty acid margaric acid, which has been shown to modulate the gating properties of PIEZO1. In brief, while we did not observe a significant blade expansion, we did observe a significant decrease in the magnitude of blade displacement (Extended Data Fig. 6).

We have extensively re-worked our discussion section to explore these physical predictions of dome curvature in more depth, with a particular focus on the protein bending stiffness and blade flexibility. We have also expanded our discussion of blade flexibility in the main text, with better reference to these physical models. However, we have kept the discussion short and simple for a lay audience of biologists.

2. Do the collected data allow to analyze the asymmetry/cooperativity of blade extensions? For this, the authors could analyze inter-blade (really: inter-fluorophore) distances for each individual channel, then calculate individual inter-blade distance ratios, and finally histograms of ratios for all channels. Inter-blade distance ratios of ~ 1 would indicate a high degree of cooperativity, whereas values deviating from 1 inform to what extent blades can move independently.

Our data does allow for a basic analysis of blade extension cooperativity. As suggested, we isolated the inter-blade distances for each identified triple-labeled channel and plotted a histogram of the ratio of the shortest, middle, and longest edges of the resultant triangle to assess the extent of cooperativity (Extended Data Fig. 7). At position 103 in the absence

of a stimulus, we observed median ratios of 0.72, 0.61, and 0.85 for the short/middle, short/long, and middle/long edges, respectively (Extended Data Fig. 7a). These values indicate that the blades do not display a high degree of cooperativity. Interestingly, stretching of the blades from lateral membrane tension induced by hypotonic shock did not significantly alter the median inter-blade distance ratios (Extended Data Fig. 7b), suggesting that cooperativity does not increase during channel gating. Overall, these data suggest that the distal blades appear to move relatively independently. We also measured inter-blade distance ratios at position 670 (Extended Data Fig. 7c) and observed an average increase in each case relative to position 103. It is possible that this increase is not driven by cooperativity, but rather the overall decrease in flexibility at this position relative to the distal blade. Additionally, we cannot discount that any degree of apparent cooperativity might be driven by other factors, such as local differences in membrane topology.

We now reference blade extension cooperativity in the discussion section.

3. The authors suggest that the varying blade flexibility is due to differential binding energies between individual blade domains. The authors may be able to test this hypothesis experimentally by altering the binding strength with point mutations (weakening) or double-cysteine crosslinking (strengthening). The latter may even allow for acute and reversible manipulation, which would be an elegant extension of their work.

We agree this would be an elegant follow up, however we believe these experiments are beyond the scope of the current work given the extensive validation and testing required. The addition of a single disulfide crosslink between PIEZO domains should increase the overall domain interface binding energy by approximately -4 kcal mol^{-1} (ref⁵). To equalize the total predicted binding energy of PIEZO repeat I (containing amino acid position 103) to that of PIEZO repeat F (containing amino acid position 670) would require an increase of $\sim 24.5 \text{ kcal mol}^{-1}$, or roughly the equivalent of 6 disulfide bonds. The requirement for structure-guided mutagenesis of 12 total cysteine residues is further complicated by the lack of empirical atomic structures of the distal blades of PIEZO1. Importantly, such experiments would require, in our opinion, extensive functional validation of each construct with cell surface staining and electrophysiology in addition to MINFLUX imaging. Overall, we expect an independent study that thoroughly examines inter-repeat binding energy will yield interesting insights into the basic mechanics of PIEZO blade function.

We now suggest this experiment as a future direction in the discussion section.

4. Line 131: An alternative mechanism for the observed heterogeneity in blade expansion may be local (nm scale) differences in membrane properties, such as topography and bending modulus. After all, these measurements are performed in cells. The authors should consider adding this interpretation to their manuscript.

We now include a reference to this alternative mechanism in the manuscript.

5. Line 328: The claim that conformation dynamics of individual membrane proteins have not been previously observed neglects all prior work using single-molecule FRET (for example <https://doi.org/10.1038/s41589-019-0240-7>). The authors better tone down their statement accordingly.

We now tone down and clarify this statement.

6. Line 101: I find the expression “roughly in plane with the plasma membrane” confusing, because the membrane is predicted to be curved into the footprint. Please try improving the clarity.

We have removed this phrase to improve clarity.

7. Figures 1A and S1A: the alpha-fold Piezo subunits are colored as if they were not domain swapped. Please correct.

The AlphaFold II prediction of PIEZO1 models only a single protomer. Since there is high flexibility between the cap and the rest of the PIEZO protein, the AFII algorithm doesn't correctly position the cap subdomain with domain swapping without the other two protomers present. We generated the trimeric AlphaFold II model by superposing E2JF22 protomer onto the three PIEZO1 subunits of PDB 6B3R. To avoid confusing the reader, we have now removed the cap (CED) in the images of structures in this manuscript.

We now additionally include the methods subsection “Structural models” that explains how we prepared the trimeric AlphaFold II model.

8. Figures S3a, b, and e: The y-axes should be rescaled to improve data visualization.

The y-axes have been re-scaled.

9. Figures 3e, f, and 4 a, b, c, d: the figure legends do not match the order of figure panels.

The order of the figure legends is now corrected.

Referee #2 (Remarks to the Author):

The manuscript by Mulhall et al. describes an in-depth analysis of mechanical gating properties in cellular environment of the Piezo1 channel containing genetically encoded fluorescence. By using nanoscopic fluorescence imaging and applying stimuli to cells the authors could directly observe the conformational changes of the Piezo1 channel blades and measure how blade expansion underlies channel activation. They found that the cellular environment can shape the structure of the Piezo1 expanding blades and show how blade expansion affects the channel gating and structure indicating the importance of the local membrane environment for determination of the structural properties of the channel. Compared to structural models of the isolated Piezo1 protein, single-molecule imaging of the channel in cellular environment enabled the authors to detect a dynamic of local protein conformations, which led them to conclude that subunits in membrane proteins could be conformationally heterogeneous. This is an interesting conclusion suggesting that improved labeling methods could further advance the research aiming to resolve the structure of membrane proteins in their native cellular environment. Overall, this study presents an important exploration of the structural dynamics of the Piezo1 channel protein. The manuscript is well written, and the authors present their results concisely. Nevertheless, the authors should consider and address the following comments that can help to significantly strengthen the interpretation of the results in this otherwise very worthwhile study.

Major:

1. Line 130-131: “... the standard deviation of inter-blade distances is greater than the experimental localization error, supporting the hypothesis that the spread of distances is driven by intrinsic blade flexibility.” Given the standard deviations of ± 4.0 nm and ± 6.0 nm for the inter-blade distance change from detergent to membrane-bound form (line 916) and cell membrane exposed to GsMTx-4 to AlphaFold II structure (line 920), respectively, compared to the median fluorophore localization error in X, Y, and Z of 5.9, 5.8, and 5.3 nm, this hypothesis does not seem to be well supported by the data shown. To strengthen this hypothesis the authors should perform additional MINFLUX imaging experiments of Piezo1 in cells treated with polyunsaturated fatty acids and margaric acid, which decrease and increase membrane stiffness, respectively (Romero et al. (2019) Nat. Commun. 10: 1200) and determine the inter-blade distances under these conditions that will give them a greater opportunity for observing differences in intrinsic blade flexibility. These experiments would also help to support their hypothesis that the plasma membrane exerts sufficient force to bend the blades of PIEZO1 (line 107-109).

We agree that it is possible that the spread of the data in the GsMTx-4 and detergent-solubilized conditions are driven by fluorophore localization error, but data throughout the manuscript does not appear to support that hypothesis. These experiments were performed to assess the extent of membrane bending stress, which manifests in a statistically significant change in the cumulative distribution of inter-blade distances as assessed by the K-S test (Fig. 2f-g). Care was taken to perform the fluorophore labeling, fixation, imaging, and analysis as identically as possible for each condition so that changes in inter-blade distances could be more accurately compared. This is reflected by nearly all median GMM fit error for individual fluorophores being consistent within ~ 1 nm for all experimental conditions. However, several of the inter-blade distance datasets in this manuscript have a standard deviation less than the median GMM fit error. For example, the standard deviation of average inter-blade distance for the detergent-solubilized condition is ± 4.0 nm (Fig. 2f), the TCO*K 670 condition is ± 4.4 nm (Fig. 3), and the TCO*K 670 + Yoda1 condition ± 2.9 nm (Fig. 4). Given the high flexibility of the distal blade (Fig. 3), we argue that the standard deviation of inter-blade distances for this position is largely driven by flexibility.

Nevertheless, we agree that performing additional MINFLUX imaging experiments of Piezo1 in cells treated with polyunsaturated fatty acids and margaric acid (MA) to alter membrane stiffness is an excellent opportunity to observe differences in apparent blade flexibility and to provide overall support our interpretation. This is also an interesting exploration of the membrane stiffness parameter in physical models created by Haselwandter and MacKinnon (please see comment and response #1 from reviewer #1). We first enriched the plasma membrane with MA according to the protocol described in Romero et al., 2019 (please see additional methods in manuscript). Briefly, the cell culture media was supplemented with 300 μ M MA 6 hours post-transfection and 18 hours before labeling. We chose 300 μ M since this concentration resulted in the largest elevation in displacement thresholds as measured by Romero, et al. with patch clamp electrophysiology. We performed fluorophore labeling, fixation, and imaging exactly as for the TCO*K 103 unenriched condition (Fig. 1 and Fig. 2e). In our analysis, we found the overall spread of conformational states to be significantly lower in the MA enriched condition compared with the unenriched control, using the F-test of the equality of two variances ($p = 0.0056$) (Extended Data Fig. 6b). The average Gaussian mixture model (GMM) fit error of localizations for the margaric acid enriched condition was only 0.92 nm less on average in XYZ compared to the unenriched condition (Extended Data Fig. 6c), indicating that variability in data acquisition is not driving the change in data variance. Overall, these data suggest that increasing membrane stiffness with a saturated fatty acid also results in an apparent decrease in effective blade flexibility. Such a decrease in fluctuations is expected from mathematical modeling of the plasma membrane, where increased membrane stiffness decreases the magnitude of random thermal fluctuations.

These mathematical models also predict that an increase in membrane bending stiffness should also flatten PIEZO1 and increase the inter-blade distances. However, we did not observe a significant change in the average inter-blade distance ($p = 0.5713$, $D = 0.2013$, KS-test) (Extended Data Fig. 6a). Increasing membrane stiffness should also lead to channel activation at smaller values of membrane tension (see <https://elifesciences.org/articles/33660>, Fig. 7). Yet, Romero et al. observes the opposite effect (an elevation of the apparent gating threshold) and another group examining the effect of MA on PIEZO2 sees no effect at all⁶. Together, these data suggest that the effect of fatty acid composition in the plasma membrane on PIEZO1 is likely more nuanced than predicted from modeling alone and may involve direct protein binding and modulation. We now include a discussion of the MA enrichment data in the main text.

We also attempted enrichment of the plasma membrane with the polyunsaturated fatty acids ω -3 PUFA eicosapentaenoic acid (EPA) according to the protocol described in Romero et al., 2019 to see whether this might increase the magnitude of blade fluctuations as predicted by membrane modeling. Unfortunately, we found that this type of enrichment was incompatible with our experimental conditions. All imaging experiments in this manuscript were performed with suspension-adapted HEK293 cells grown under shaking conditions to keep the media aerated and in a single-cell suspension. These culture conditions are critical for washing away excess fluorophores during labeling, which would otherwise result in unacceptable levels of background on the coverslip and consequently obscure imaging of the plasma membrane. Yet, despite testing multiple enrichment concentrations within the ranges shown to produce effects on PIEZO channel gating, we found that EPA enrichment resulted in nearly complete cell death over the course of 8 hours. We suspected that increased membrane fluidity from EPA enrichment also increased the fragility of the cells, which would be especially detrimental under conditions of shaking. Therefore, we also tried several modified static culture conditions in which the cells were plated onto petri dishes and cultured without shaking. While >95% of unenriched cells survived under these modified conditions, again, nearly all the cells in the enriched condition were killed as assayed by a trypan blue exclusion test. We therefore suspect that cell death is being induced by interaction of EPA with specific components of the special cell culture medium required for shaking, the composition of which is proprietary. While it is possible that altering the media formulation or switching to a new cell line might improve cell survival, different cell culture media formulations can induce alterations in membrane composition, and different cell lines have intrinsic differences in membrane composition. Therefore, these altered conditions would not be a 1:1 comparison to the data in this manuscript.

2. Line 156-165: The authors suggest that GsMTx-4 inhibits channel activity by compacting the blades of Piezo1 resulting from reduction of membrane bending stress due to GsMTx-4 interaction with the lipid bilayer acting as a “mobile reserve” of membrane material. However, GsMTx-4 penetration into the membrane bilayer shifts between shallow and deep penetration depending on bilayer tension (Gnanasambandam et al. Biophys. J. 112(1): 31-45, 2017), which distorts the distribution of membrane tension and thus, to a different extent, affects the transfer of force from the bilayer to an inherently mechanosensitive ion channel like Piezo1. Since the standard deviation for the inter-blade distance change from cell membrane exposed to GsMTx-4 to AlphaFold II structure was ± 6.0 nm and thus it is within the range of the median fluorophore localization error (see the previous comment), the concentration of 20 μ M GsMTx-4 applied in MINFLUX cell experiments might not have been adequate to detect the effect suggested by the cartoon shown in Fig. 2g. MINFLUX

imaging of Piezo1 reconstituted into a well-defined lipid environment of lipid vesicles (Lin et al. (2019) Nature 573: 230-234) would allow for better distinction of the GsMTx-4 effect on Piezo1 blade extension by applying it at low and high concentrations. Although desirable, I leave it to the authors to decide if they wish to include these experiments in the revised manuscript. They should, nevertheless, comment on the appropriateness of choosing the concentration of 20 μM GsMTx-4 in their study.

For the GsMTx-4 condition specifically, we suspect that the standard deviation is driven by two primary factors: the inherent flexibility of the blades in the plasma membrane and the identification of triple-labeled PIEZO1 proteins which do not have a high local concentration of GsMTx-4. GsMTx-4 has an equilibrium binding constant for the lipid bilayer of 2 μM and unbinds from the membrane very quickly ($k_{\text{off}} \approx 0.2 \text{ s}^{-1}$). Therefore, after removing the GsMTx-4 solution, $\sim 63.2\%$ of the total bound GsMTx-4 is in theory expected to unbind after just 5 seconds. For that reason, we used 20 μM – a concentration 10 times the equilibrium binding concentration for the plasma membrane – to saturate the plasma membrane as much as possible and obtain the largest effect possible. We also applied fixative very quickly after washing off the GsMTx-4 to capture a state in which the most GsMTx-4 is bound to the membrane. This resulted in a statistically significant decrease in inter-blade distances by the K-S test (Fig. 2g). We now clarify our use of 20 μM GsMTx-4 in the main text.

We agree that the overall mechanism of GsMTx-4 appears to involve shifting between shallow and deep penetration depending on bilayer tension, and that this affects the transfer of force to PIEZO1 via acting as a “mobile reserve” of membrane material. We now include a reference to this mechanism in the main text. Yet, it appears from our data that the peptide also inserts into general distortions of the membrane that do not arise from lateral tension alone, such as from the bending stress that expand the blades of PIEZO1 in a planar lipid bilayer. Dissecting this effect in much more detail in the well-defined environment of purified PIEZO1 protein in a homogenous lipid bilayer would be interesting. However, observing this effect would likely require reconstitution into a planar bilayer rather than vesicles so that the membrane bending forces are sufficient to be reversed by application of GsMTx-4. Additional experimental hurdles come from the requirement of immobilizing the protein onto the surface to limit diffusion over the long timescales required for MINFLUX imaging, and supporting the lipid bilayer onto the glass coverslip in a way that does not result in protein physisorption onto the supporting substrate which can alter the conformation of the protein. For these technical challenges, we believe these experiments to be beyond the scope of this current paper, although we are excited about these as future directions.

3. Line 178: “PIEZO1 is spontaneously active in the absence of lateral membrane tension in a cell ...”. Spontaneous activity of Piezo1 has been observed in membrane patches in patch clamp experiments and therefore, this is not necessarily true for a cell. This is because the resting tension in the dome of a membrane patch ranges from 1–4 pN/nm (mN/m) depending on the recording mode (Suchyna et al. (2009) Biophys. J. 97(3): 738-47), which is sufficient to induce spontaneous activity of Piezo1.

We apologize for the confusion. Citations (#38-39 in the submitted manuscript, #54-55 in the revised manuscript) for this sentence measure resting open probability in membrane patches during a 5 mm Hg pre-pulse at -80 mV, which is designed to flatten the membrane within the membrane patch and minimize membrane creep. This creates a condition in which the membrane experiences a nominal absence of membrane tension. The degree of spontaneous activity measured in these experiments is approximately 0.5%. We have clarified this sentence, which now resides in the discussion section.

Citation 40 was meant to show that spontaneous activity has been measured in artificial asymmetric lipid bilayers containing reconstituted PIEZO1 protein. While some of these experiments are thought to lack lateral membrane tension, we cannot discount that other forces – not present in an unperturbed, roughly planar lipid bilayer – are driving such activity. We have therefore removed this reference from this sentence.

Minor:

1. Line 136: “ ... blade expansion in a cell is due ‘to’ membrane bending forces.”

This error has been corrected.

2. Line 175-176: “ ... exerted primarily through the plasma membrane rather than through tethering to the cytoskeleton or extracellular matrix.” The papers by Cox et al. (Nat. Commun. 7: 10366, 2016) and Syeda et al. (Cell Rep. 17: 1739–1746, 2016) reporting first the activation of Piezo1 by forces from the plasma membrane should be cited here.

The relevant citations have been added.

3. Line 240: "... in response 'to' hypo-osmotic shock."

This error has been corrected.

4. Line 328-329: "The data described here show, to our knowledge, the first time that conformational dynamics of individual membrane proteins have been directly observed with fluorescence microscopy at the level of single molecules." This statement is incorrect. MINFLUX fluorescence imaging of Nup96, a nuclear pore complex protein was done previously as reported in reference 31 (Schmidt et al. (2021) Nat Commun 12: 1478).

The Schmidt et al. (2021) Nat Comm 12: 1478 reference examines the structure of the nuclear pore complex, a large and heterogeneous complex of membrane proteins, with MINFLUX imaging. However, no conformational changes or dynamics were claimed or observed in this citation. We now further clarify and tone down this statement, also incorporating suggestions from reviewers #1 and #3.

5. Fig. 1d, line 852-853: Blue and yellow clusters on the bottom right, do not match the blue and yellow clusters on the top after 90o rotation.

Fig. 1d, bottom right, now shows the correct rotation.

Referee #3 (Remarks to the Author):

Do I feel the results presented are of immediate interest to many people in my own discipline or to people from several disciplines?

Yes, for a very interesting structural investigation applying these types of microscopy and analysis on a complex outside of the standard samples used for technique development. I think this is the first example in an original biological experiment of observing conformational states and deducing other properties using this type of microscopy, which can access states more native than is possible in single-particle cryo-EM.

More clarity is needed about the analysis, and therefore the results and their interpretation. If these can be improved, publication of this manuscript will advance the use of these techniques and improve the understanding of their potential.

Statistics: The authors have checked n/a for null-hypothesis testing, but have use K-S tests with significance values. The checklist indicates that the test statistic (often D) should be included in the presentation of results, but is missing.

The D statistic is now included in the figure legends for each K-S test, and the appropriate test statistic is now included for other statistical tests.

Some uncertainties on measurements and comparisons have been included, but more would be helpful.

We have added additional statistical tests and uncertainties to our manuscript: Fig. 1e, Fig. 3d, Fig. 3f – bottom, Extended Data Fig. 5, Extended Data Fig. 6b, and Extended Data Fig. 6d – bottom.

Abstract

I am not clear on how surprising it is that expansion should vary along the blade, especially with greater flexibility further from the centre.

We have added an additional discussion in the main text exploring this possibility. We also now discuss overall blade flexibility in more detail in the discussion section.

These methods will serve as a framework for structural analysis of membrane proteins...

The application of these methods in an experiment like this is indeed an advance, but the methods themselves are each not a novel development, although bespoke elements have been used in the analysis.

We now clarify this statement.

Segmentation and particle fusion

...promoting a 3-fold symmetry by rotating each initially aligned particle by a random integer factor of $2\pi/3$...

It should be noted that the very clear 3-fold symmetry observed in the super-particle of Fig. 1c and used in other iPALM results is ensured by this "symmetry promotion", if particles do share a preferred orientation to align on (but they do not need to be 3-fold symmetric). It is illustrated in Ext. Data Fig. 4d, but this "symmetry-promoted" term should be used in the text and Fig. 1c (and possibly elsewhere) to avoid the reader thinking that a super-particle without this symmetry promotion would have this appearance.

We now make it clear in the manuscript and in the legend of Fig. 1 that the super-particle is has 3-fold symmetry promotion.

It would be good to consider the effect of this symmetry promotion on the elongated density patterns that are noted in the super-particle.

A visualisation of the super-particle before symmetry promotion (perhaps similar to Fig. 1c) would be informative.

We now include Extended Data Fig. 5, a supplemental figure comparing a super-particle without symmetry promotion to one with 3-fold symmetry promotion. The reason that symmetry promotion is necessary is that the number of detected localizations per fluorophore position is not uniform, and the registration algorithm tends to match dense regions of localizations. This results in "hot spots" of localizations that are preferentially fused together^{7,8}. The PIEZO1 super-particle without symmetry enforcement results in a roughly trimeric shape (Extended Data Fig. 5b, left), but the preference to match dense regions results in two major hot spots results in an unbalanced super-particle. Encouragingly, the measured distance between the two main hot spots (25.0 ± 2.4 nm) is nearly the same as for the symmetry-enforced super-particle (25.4 ± 5.9 nm).

Given this bias on the super-particle, I am not clear on the significance of the inter-blade distances measured within it. These are distances between localisation positions in rotated particles in $\sim 2/3$ cases, not measurements between localisations in aligned particles without symmetry promotion. In particular, the effect on uncertainties of these measurements within the symmetry-promoted particle needs considering. Further explanation of the use of these measurements in this light would be helpful.

Using intra-complex pair-wise distances between localisations or clusters without alignment and averaging may be an alternative.

We agree that accurately measuring inter-blade distances from the symmetry-promoted super-particle is difficult. Of largest concern is that the inter-blade distances are very near that of the iPALM resolution (10-20 nm), discounting the error propagated from sources such as drift and vibration. We view this data to be somewhat qualitative in nature, although indicative that something more interesting might be happening. We are certainly pushing this method at or beyond its limit, especially for flexible proteins which do not form a roughly isotropic cloud of localization density. These are the reasons why single-molecule measurements with MINFLUX are so important.

As suggested, we now measure inter-density pairwise distances for the two prominent localizations clouds which could be clustered in the super-particle without symmetry enforcement (Extended Data Fig. 5).

Single-molecule imaging of Piezo1 in a cell membrane

...at least several nanometres more extended...

Perhaps 'at least' is not needed here. 'Most probable' appears right (given the caveats above) from Fig. 1e, but an uncertainty value would be helpful (also considering the caveats above). The authors could consider the use of the functions in Churchman (Biophys. J. 90, 668-671 (2006)) for estimating mean and variability.

We now measure mean and variability calculated directly from the pairwise distances and include these in Fig. 1e and the figure legend. As noted above, we also measured these distances from the super-particle without symmetry enforcement (Extended Data Fig. 5b).

MINFLUX data acquisition

Laser powers are given as percentages, but this then needs more information about the lasers (e.g. what would maximum power have been?)

The laser power in the sample plane per % set power in the software for the excitation and activation lasers are now noted in the methods. We also provide an additional clarification about power ramping during the targeting iteration.

'Mean standard deviation for stabilization' needs explaining. And is then corrected for in some way?

We now provide an abbreviated overview of stabilization and the stabilization error.

MINFLUX data analysis

...the first DBSCAN...required 5 neighbours for the core point...

This should be ...for 'a' core point...

This error has been corrected.

'candidate clusters containing blade angles > 120 degrees were filtered out to eliminate non-specific trace streaks'
What are these streaks? Is natural variability of the complexes actually being removed?

These streaks are thick regions of localizations perpendicular to the sample plane that often consist of several traces. We do not know the origin of these streaks, but they are present on our instrument in diverse types of samples that do not have PIEZOs, including reference nuclear pore complexes labeled with NUP96-SNAP (ref⁹). Given their ubiquity amongst samples in our hands, we feel confident that they are not labeled PIEZOs specifically. We are currently working with Abberior Instruments to expose the origin of these streaks.

It is better in the case of the MINFLUX data that the centers of the clusters are used for direct intra-complex distance measurements.

What are the centre positions for a cluster? Centroid? Centre of mass?

Each cluster is first identified using the DBSCAN algorithm and is then individually subjected to a 3D Gaussian mixture model (GMM) fit (fitgmdist in MATLAB). The mean of the GMM fit is reported as the center position of the cluster. We now note in the methods that "The fluorophore center positions are estimated as the mean values of the GMM fit".

Blade flexibility is driven by a differential energetic stability

'Notably, the s.d. of inter-balde distances was 1.6 times less at position 670...'

For this to appear notable, more explanation might be needed. Position 670 is closer to the fixed centre than position 103, and a smaller variation of its position relative to the centre might be expected, even if each repeat has similar flexibility. - Or those compounded flexibilities along the blade might be expected to result in a wider range of positions at position 103.

We have added an additional discussion in the main text exploring the possibility that the mechanical properties of the blade might be uniform.

There are no uncertainties on the bending distances: 'the blade at proximal repeat F is bent by only 2.8 nm, compared to 6.1 nm at distal repeat.'

This affects the significance of this difference, for which an uncertainty value would also help.

For clarity, we now specify these as average changes in distance in both the main text and the figure legends. Since these are simply comparisons with just a single value from the structure, we are not able to run accurate statistical tests. Since the spread of the experimental values seems to be driven by molecule flexibility rather than measurement error, we found it best to leave these as average values to avoid confusing the reader.

The panel letters in Fig. 3 do not match the caption.

The panel letters are now corrected.

The illustrations in Fig. 3f are not very informative. A density plot showing the structure smeared out by the range of angles might be more informative.

We have removed the illustrations, and instead now include a statistical test of the angles in Fig. 3f, bottom.

"The blades were on average more symmetrical" at repeat F vs. repeat I.

The variabilities are large, and a statistical assessment of the significance of this difference would be helpful.

We now perform a statistical test of the variation in Fig. 3d.

Activation of Piezo1 by blade extension

Projected area is used only for the test of osmotic shock response of Piezo1, not inter-blade distance, as is used everywhere else. Why is this?

For the hypo-osmotic swelling condition we measured inter-blade distance (Fig. 4b) and calculated the projected area from the circumradius of the triangle formed by the inter-blade distances (Fig. 4c). Based on the previous use of projected area as a metric for blade expansion and the mechanics of channel gating in the literature^{1-3,10-12}, we determined it informative for the field to also include it in the experimental condition where blade expansion is greatest. We suspect that such data might be useful to other groups for refining previous models of channel gating.

The panel letters in Fig. 4 do not match the caption.

The panel letters are now corrected.

'Distances are shown' should be 'Areas are shown' at one point in Fig. 4.

This is now corrected.

...position 103 ... incubated with ... Yoda1, the average inter-blade distances increased by 2.08 nm (Fig. 4f).

This ideally needs an uncertainty value and the statistical information ($p=0.393$) should be brought closer to it.

This is now corrected.

The corresponding result for position 670, 1.96 nm, would also ideally have an uncertainty on it.

We now say "on average" to clarify that these are average changes in position.

In the light of the weak $p=0.393$ result for position 103 and the absence of uncertainties on the measurements, the assertion that Yoda1 uniformly expands the blade seems a bit strong, currently.

We now tone down and clarify this statement in the main text.

Also perhaps the highlight of the power of the technique to accurately observe nm-scale molecular movements - this does seem impressive at times in the manuscript, but the inconclusive measurements of position 670 in this case do not show that.

We contend that the expansion of position 670 by Yoda1 is conclusive. The change in position is statistically significant relative to the -Yoda1 condition ($p = 0.0464$ and $D = 0.38$, Kolmogorov-Smirnov test). The change in distance also is nearly the same as for position 103 (average $\Delta 1.96$ nm at position 670 vs average $\Delta 2.08$ nm at position 103) which would be expected from Yoda1 “wedging” into the binding pocket between proximal repeats repeat A and B¹³. However, the change in position 103 from Yoda1 binding is not statistically significant simply because of the inherent flexibility of these domains.

Discussion

'The extent of blade expansion is driven by a gradient of blade flexibility' - Such a gradient does not seem clear from the imaging, as opposed to possibly constant flexibility along the blade resulting in greater movement at distal regions. It may arise from the binding energy calculation, though.

We now provide an extended discussion of blade flexibility in both the main text and the discussion sections that clarify these statements, and tone down these statements.

'We also show that the extent of Piezo1 activation... appears to be explained by the degree of blade expansion.'
Is the data more tentative than this? The authors did put forward a theory on why the degree of channel activation appeared similar for chemical and mechanical modulators, but not one that was covered by the experiments.

We have toned down this statement accordingly.

'Correlation between the extent of blade extension and channel activity'
The similarity of activations has been presented results, but not so much this correlation at different blade extension.

We argue that our data shows that the extent of channel activation appears to correlate with channel activity. Our data is supported by both extensive physical models of PIEZO1 activation and electrophysiological studies. We now include an extended discussion regarding these physical models and their relation to blade expansion and channel activity.

'We show how the differential flexibility of the blades is determined by the binding strength of Piezo repeat domains.'
This currently seems a circular argument, since the imaging does not seem conclusive about a flexibility gradient (as opposed to constant but compounded flexibility along the blade), and the binding strengths are currently the clearest values suggested to produce such a gradient.

We now tone down this statement and include an extended discussion of blade flexibility.

'These properties are likely dictated by graded blade flexibility.'
Again, why would the flexibility need to be graded for this property?

We have removed this sentence in our revamped discussion section that more thoroughly addresses blade flexibility.

'We expect that these methods will provide a foundation for the use of fluorescence nanoscopy for single-molecule structural biology.'

Yes, iPALM, MINFLUX and Heydarian's particle alignment (albeit with the symmetry promotion) are the foundation. And this is an exciting application where these techniques have been brought to bear on a genuine biological question. If they can be clearer about the results (given the analysis methods) and their interpretation, then the authors have shown the potential of these techniques in original structural biological experiments for the first time, to my knowledge.

References

- 1 Haselwandter, C. A., Guo, Y. R., Fu, Z. & MacKinnon, R. Elastic properties and shape of the Piezo dome underlying its mechanosensory function. *Proc Natl Acad Sci U S A* **119**, e2208034119 (2022). <https://doi.org:10.1073/pnas.2208034119>
- 2 Haselwandter, C. A., Guo, Y. R., Fu, Z. & MacKinnon, R. Quantitative prediction and measurement of Piezo's membrane footprint. *Proc Natl Acad Sci U S A* **119**, e2208027119 (2022). <https://doi.org:10.1073/pnas.2208027119>
- 3 Guo, Y. R. & MacKinnon, R. Structure-based membrane dome mechanism for Piezo mechanosensitivity. *Elife* **6** (2017). <https://doi.org:10.7554/eLife.33660>
- 4 Tyler, A. I. I., Greenfield, J. L., Seddon, J. M., Brooks, N. J. & Purushothaman, S. Coupling Phase Behavior of Fatty Acid Containing Membranes to Membrane Bio-Mechanics. *Front Cell Dev Biol* **7**, 187 (2019). <https://doi.org:10.3389/fcell.2019.00187>
- 5 Krissinel, E. & Henrick, K. Inference of macromolecular assemblies from crystalline state. *J Mol Biol* **372**, 774-797 (2007). <https://doi.org:10.1016/j.jmb.2007.05.022>
- 6 Zheng, W., Nikolaev, Y. A., Gracheva, E. O. & Bagriantsev, S. N. Piezo2 integrates mechanical and thermal cues in vertebrate mechanoreceptors. *Proc Natl Acad Sci U S A* **116**, 17547-17555 (2019). <https://doi.org:10.1073/pnas.1910213116>
- 7 Heydarian, H. *et al.* 3D particle averaging and detection of macromolecular symmetry in localization microscopy. *Nat Commun* **12**, 2847 (2021). <https://doi.org:10.1038/s41467-021-22006-5>
- 8 Heydarian, H. *et al.* Template-free 2D particle fusion in localization microscopy. *Nat Methods* **15**, 781-784 (2018). <https://doi.org:10.1038/s41592-018-0136-6>
- 9 Thevathasan, J. V. *et al.* Nuclear pores as versatile reference standards for quantitative superresolution microscopy. *Nat Methods* **16**, 1045-1053 (2019). <https://doi.org:10.1038/s41592-019-0574-9>
- 10 De Vecchis, D., Beech, D. J. & Kalli, A. C. Molecular dynamics simulations of Piezo1 channel opening by increases in membrane tension. *Biophys J* **120**, 1510-1521 (2021). <https://doi.org:10.1016/j.bpj.2021.02.006>
- 11 Lin, Y. C. *et al.* Force-induced conformational changes in PIEZO1. *Nature* **573**, 230-234 (2019). <https://doi.org:10.1038/s41586-019-1499-2>
- 12 Wang, L. *et al.* Structure and mechanogating of the mammalian tactile channel PIEZO2. *Nature* **573**, 225-229 (2019). <https://doi.org:10.1038/s41586-019-1505-8>
- 13 Botello-Smith, W. M. *et al.* A mechanism for the activation of the mechanosensitive Piezo1 channel by the small molecule Yoda1. *Nat Commun* **10**, 4503 (2019). <https://doi.org:10.1038/s41467-019-12501-1>

Reviewer Reports on the First Revision:

Referees' comments:

Referee #1 (Remarks to the Author):

All my suggestions were well addressed and I am very happy with the manuscript.

Referee #2 (Remarks to the Author):

In the updated edition of their manuscript, the authors have adequately addressed my remarks. Alongside their responses to the feedback provided by the other two reviewers, this has greatly enhanced and solidified the study as described in the revised manuscript.

Referee #3 (Remarks to the Author):

The manuscript is improved. I recommend these points should still be addressed:

Uncertainties

The uncertainties on results still need adding in places in the text for interpretation of the results.

The uncertainties on results still need adding in places in the text.

. l. 77 : 19.2 nm, as in Ext. Data Fig. 5, would be better than ~19 nm.

.. Currently, the SD bars in Fig. 1e and Ext. Data Fig. 5a extend below the level of the AlphaFold result, even though $25.4 - 5.9 > 19.2$.

. l. 109 : 6.2 ± 5.9 nm (although see discussion of symmetry promotion and results below)

. l. 210 : The sample size in the KS-test will help to understand the p-value result.

. l. 222 and Fig. 3c legend : The uncertainties on the expansion distances are available and useful.

. l. 267-8 : Both values and uncertainties are needed for good interpretation, not just one of them and the difference. Giving the sample size will help understand the K-S test result.

. l. 289 (2.08 nm): Both values and uncertainties (with and without Yoda1) are needed for good interpretation, or more simply, the uncertainty on their difference from error propagation. Giving the sample size will help understand the K-S test result.

. l. 291 (1.96 nm): Same comment as for l. 289.

Symmetry promotion, Fig, 1c-e and Ext. Data Fig 5

I agree that with the authors that in the end, the symmetry promoted result is somewhat qualitative. Therefore, the treatment of the 'elongation' of the clusters and the inter-blade measurement in the symmetry-promoted super-particle should be treated cautiously. Currently quantitative results are given from this qualitative result without these caveats, which should at the least be addressed, as well as the statement of symmetry promotion.

The Ext. Data Fig. 7 is also very informative about this (and a good addition), with the complex not in general being symmetric.

The results in Ext. Data Fig. 5 are quantitative results that merit presenting to the reader as more important than those from the symmetry-promoted super-particle, in my opinion, even though they look messier. They possibly include an overall asymmetry of the particle, if the third dense region is considered, in agreement with Ext. Data Fig. 7, and allow a much less biased measurement of the distance between position 103 in two blades.

I would prefer to see the non-symmetry promoted results presented as the key data, with the symmetry-promoted super-particle included as a qualitative illustration of an idealised visualisation of the data for qualitative consideration, but notably revealing the same dimensions.

This would agree more clearly with the real individual particles shown in Ext. Data 4c as well.

Laser power

The power information is nice. I wondered whether there was any more detailed product information that future experimentalists needed, or whether these were just the standard lasers that came with the MINFLUX.

Binding energies and flexibility discussion

I feel the treatment of these calculations is improved in the manuscript. I would suggest moving the calculations to after the MINFLUX results, as part of discussing their possible implications, rather than beginning with them in the flexibility analysis section.

Fig. 3f, Ext. Data Fig. 6d and text

A decrease in a Gaussian R2 values does not mean the distribution becomes less symmetric, but less well-explained by a Gaussian curve. e.g. it could actually be more symmetric, but have a different shape to the peak. It is not clear what numbers the lower symmetry argument is based on, from the fit results.

The fit results are not commented on in an equivalent way in Ext. Data Fig. 6d.

Also, the test in Fig. 3f/Ext. Data Fig. 6d bottom is ok, I think, but quite indirect. As it stands, more information should be given on the deviations from 60 degrees. Probably medians (and e.g. quartiles on the plot) for these highly-skewed distributions with the current test.

To take care of all of this, would comparing the two inter-blade angle distributions with an F-test for a difference in their variances help (assuming they pass a normality test - or non-parametric equivalent)? This might quantify the point intuitively.

Fig. 3d and Ext. Data Fig. 6b

Where do the confidence intervals for the variances come from? I am not sure they are needed as well as the numerical statistical test result.

Flexibility similarity at positions 103 and 670

. l. 292: A difference of within 0.12 is quoted, but again, this will have quite some uncertainty on it, which could be calculated with error propagation. That seems needed to present and help with interpreting this claim.

Discussion

. Binding energies, l. 321: 'These calculations' seems more appropriate than 'These observations', for presenting this theoretical aspect of the work.

Details

. Extended data Fig. 7: Ratios are labelled as Distance (nm)

. l. 152-3 : 'segmentation algorithm' : The authors do more than segmentation - perhaps something like 'analysis' or 'analysis pipeline'?

. l. 163: Something like 'consistent with', rather than 'indicates that', seems more appropriate.

. l. 212 : 'each measured blade position' : This needs to be more specific. Should this be 'between positions 103 and 670'?

. l. 226 : 'Additional evidence': This seemed to be the first time such an 'intermediate state' was clearly discussed.

. l. 257-8: This sentence did not make sense.

Author Rebuttals to First Revision:

Referees' comments:

Referee #1 (Remarks to the Author):

All my suggestions were well addressed and I am very happy with the manuscript.

Referee #2 (Remarks to the Author):

In the updated edition of their manuscript, the authors have adequately addressed my remarks. Alongside their responses to the feedback provided by the other two reviewers, this has greatly enhanced and solidified the study as described in the revised manuscript.

Referee #3 (Remarks to the Author):

The manuscript is improved. I recommend these points should still be addressed:

Uncertainties

The uncertainties on results still need adding in places in the text for interpretation of the results.

The uncertainties on results still need adding in places in the text.

. l. 77 : 19.2 nm, as in Ext. Data Fig. 5, would be better than ~19 nm.

.. Currently, the SD bars in Fig. 1e and Ext. Data Fig. 5a extend below the level of the AlphaFold result, even though $25.4 - 5.9 > 19.2$.

These panels have been updated.

. l. 109 : 6.2 ± 5.9 nm (although see discussion of symmetry promotion and results below)

We have added the associated error.

. l. 210 : The sample size in the KS-test will help to understand the p-value result.

These values are present in the associated figure legend.

. l. 222 and Fig. 3c legend : The uncertainties on the expansion distances are available and useful.

We now include these uncertainties in the main text.

. l. 267-8 : Both values and uncertainties are needed for good interpretation, not just one of them and the difference. Giving the sample size will help understand the K-S test result.

We now provide each value and uncertainty. Sample sizes are included in associated figure legend.

. l. 289 (2.08 nm): Both values and uncertainties (with and without Yoda1) are needed for good interpretation, or more simply, the uncertainty on their difference from error propagation. Giving the sample size will help understand the K-S test result.

. l. 291 (1.96 nm): Same comment as for l. 289.

For both above comments: these values are included in the associated figure legends, and we believe that repeating them in the main text is unnecessary.

Symmetry promotion, Fig, 1c-e and Ext. Data Fig 5

I agree that with the authors that in the end, the symmetry promoted result is somewhat qualitative. Therefore, the treatment of the 'elongation' of the clusters and the inter-blade measurement in the symmetry-promoted super-particle should be treated cautiously. Currently quantitative results are given from this qualitative result without these caveats, which should at the least be addressed, as well as the statement of symmetry promotion.

The Ext. Data Fig. 7 is also very informative about this (and a good addition), with the complex not in general being symmetric.

The results in Ext. Data Fig. 5 are quantitative results that merit presenting to the reader as more important than those from the symmetry-promoted super-particle, in my opinion, even though they look messier. They possibly include an overall asymmetry of the particle, if the third dense region is considered, in agreement with Ext. Data Fig. 7, and allow a much less biased measurement of the distance between position 103 in two blades.

Although we agree with the reviewer that the super-particle without symmetry promotion is important, we think that the super-particle with symmetry promotion is far more informative of domain flexibility. Additionally, reliably separating out three densities from the super-particle without symmetry promotion using K-means clustering is not possible, and thus does not capture the same amount of distance information compared with the symmetric super-particle.

I would prefer to see the non-symmetry promoted results presented as the key data, with the symmetry-promoted super-particle included as a qualitative illustration of an idealised visualisation of the data for qualitative consideration, but notably revealing the same dimensions.

We contend that since the super-particle without symmetry promotion only clusters two distinct positions, there is less meaningful information within it. Additionally, given that the registration algorithm tends to preferentially match dense regions of localizations, the non-symmetry promoted super-particle is dominated by the 'hot spots'. Ultimately, information about the degree of flexibility and the most probable distances are largely lost in the non-symmetry promoted super-particle. We do include the sentence "With prior knowledge of subunit stoichiometry, we enforced 3-fold symmetry during the creation of the super-particle since the registration algorithm tends to match regions of dense localizations^{28,29} (Extended Data Fig. 5a-b)" in the main text, which we hope makes the reason for symmetry promotion clear.

This would agree more clearly with the real individual particles shown in Ext. Data 4c as well.

Upon visual inspection, the identified single particles look similar to the super-particle without symmetry promotion. But our data suggests that these are simply snapshots of distinct conformations due to the flexible blade domains.

Laser power

The power information is nice. I wondered whether there was any more detailed product information that future experimentalists needed, or whether these were just the standard lasers that came with the MINFLUX.

These are the lasers that come standard with the commercial version of the MINFLUX microscope from Abberior Instruments. No additional modifications to the microscope were made.

Binding energies and flexibility discussion

I feel the treatment of these calculations is improved in the manuscript. I would suggest moving the calculations to after the MINFLUX results, as part of discussing their possible implications, rather than beginning with them in the flexibility analysis section.

We appreciate the reviewer's suggestion. However, we prefer the current layout.

Fig. 3f, Ext. Data Fig. 6d and text

A decrease in a Gaussian R² values does not mean the distribution becomes less symmetric, but less well-explained by a Gaussian curve. e.g. it could actually be more symmetric, but have a different shape to the peak. It is not clear what numbers the lower symmetry argument is based on, from the fit results.

We agree that our wording was not ideal. We now clarify these statements.

The fit results are not commented on in an equivalent way in Ext. Data Fig. 6d.

We now comment on each histogram equivalently.

Also, the test in Fig. 3f/Ext. Data Fig. 6d bottom is ok, I think, but quite indirect. As it stands, more information should be given on the deviations from 60 degrees. Probably medians (and e.g. quartiles on the plot) for these highly-skewed distributions with the current test.

We agree that medians are the more accurate metric to present given the statistical test used. We now present the error bars in these figures as median \pm the 95% confidence interval.

To take care of all of this, would comparing the two inter-blade angle distributions with an F-test for a difference in their variances help (assuming they pass a normality test - or non-parametric equivalent)? This might quantify the point intuitively.

While we agree that the F-test is also reasonable, we prefer the Mann-Whitney U test.

Fig. 3d and Ext. Data Fig. 6b

Where do the confidence intervals for the variances come from? I am not sure they are needed as well as the numerical statistical test result.

The confidence interval is calculated using the chi-squared distribution with a 95% confidence interval. We think they are informative and would prefer to keep them.

Flexibility similarity at positions 103 and 670

l. 292: A difference of within 0.12 is quoted, but again, this will have quite some uncertainty on it, which could be calculated with error propagation. That seems needed to present and help with interpreting this claim.

We now remove this reference and now state: "...remarkably close on average compared to the Yoda1-induced change in distance at position 103". Actual values and errors are presented in the figure legend.

Discussion

. Binding energies, l. 321: 'These calculations' seems more appropriate than 'These observations', for presenting this theoretical aspect of the work.

We have changed this to the reviewer's preferred wording.

Details

. Extended data Fig. 7: Ratios are labelled as Distance (nm)

We have made this correction.

. l. 152-3 : 'segmentation algorithm' : The authors do more than segmentation - perhaps something like 'analysis' or 'analysis pipeline'?

We now use the term 'analysis pipeline'.

. l. 163: Something like 'consistent with', rather than 'indicates that', seems more appropriate.

We have reworded this statement.

. l. 212 : 'each measured blade position' : This needs to be more specific. Should this be 'between positions 103 and 670'?

We have incorporated this suggestion.

. l. 226 : 'Additional evidence': This seemed to be the first time such an 'intermediate state' was clearly discussed.

We have clarified this sentence to refer to a "resting" rather than "intermediate" state.

. l. 257-8: This sentence did not make sense.

Fixed.